# Non-additive microbial community responses to environmental complexity

Alan R. Pacheco [1,2], Melisa L. Osborne [1,2] & Daniel Segrè [1,2,3,4,5 ✉]

Environmental composition is a major, though poorly understood, determinant of microbiome dynamics. Here we ask whether general principles govern how microbial community growth yield and diversity scale with an increasing number of environmental molecules. By assembling hundreds of synthetic consortia in vitro, we find that growth yield can remain constant or increase in a non-additive manner with environmental complexity. Conversely, taxonomic diversity is often much lower than expected. To better understand these deviations, we formulate metrics for epistatic interactions between environments and use them to compare our results to communities simulated with experimentally-parametrized consumer resource models. We find that key metabolic and ecological factors, including species similarity, degree of specialization, and metabolic interactions, modulate the observed non-additivity and govern the response of communities to combinations of resource pools. Our results demonstrate that environmental complexity alone is not sufficient for maintaining community diversity, and provide practical guidance for designing and controlling microbial ecosystems.

[1] Graduate Program in Bioinformatics, Boston University, Boston, MA, USA. [2] Biological Design Center, Boston University, Boston, MA, USA. [3] Department of Biology, Boston University, Boston, MA, USA. [4] Department of Biomedical Engineering, Boston University, Boston, MA, USA. [5] Department of Physics, Boston University, Boston, MA, USA. ✉email: dsegre@bu.edu

Microbial communities form the basis for an enormous range of biological processes, from cycling of nutrients in the ocean to regulation of human health[1–5]. Despite our growing knowledge of community compositions in various biomes[6–8] and of the role of individual nutrients in modulating community properties[9,10], relatively little is known about how the nutrient complexity of an environment (i.e., the number of different available nutrients) affects community ecology. Understanding this relationship is crucial to disentangling the effects of the environment on natural microbial ecosystems, which are exposed to a multitude of different nutrients[9,10], as well as the effects of diet on microbiome structure and function. In the gut microbiome, for example, recent work has highlighted how community composition depends strongly on the diversity of available nutrients[11–13]. However, reports in natural ecosystems[14–16] and in synthetic microcosms[17–19] conflict on how this environmental complexity modulates growth yields and taxonomic diversity, raising questions as to how these relationships vary within and across communities, and hindering nascent efforts to engineer microbiomes with defined functions[20–22].

An additional unknown in the effect of environmental complexity on community assembly is the extent to which different nutrient compositions drive ecosystems towards predictable states, as opposed to stochastically driven outcomes. While previous work has shown a combination of determinism and stochasticity in community assembly[23,24], studies have also shown how particular environments can be associated with long-lasting stable communities[25]. It is therefore important to understand to what degree these patterns will impact synthetic consortia cultured on increasingly complex combinations of defined nutrients.

A number of quantitative frameworks have previously been used to address similar questions, and have provided possible clues as to how a microbial community could depend on the complexity of its environment. In classical ecology, for example, theories based on competitive exclusion and niche partitioning suggest that there would be greater opportunities for biodiversity in environments with a greater breadth of nutrient types[26,27]. Although this is an intuitive hypothesis, factors such as organism-specific resource use capabilities[28,29], ecological niche overlap[30,31], and interspecies interactions[24,32,33] can lead to significant deviations from this expectation. From a very different perspective, the question of how different perturbations in biological systems would be expected to jointly affect a given phenotype is captured by the classical genetic concept of epistasis[34,35]. Epistasis between two genetic mutations, for example, quantifies how much the phenotypic effect of one mutation is affected by the presence of the other. This concept constitutes a broader systems biology framework for quantifying the nonlinear behavior of biological systems[36–39], and can be used to estimate the non-additivity of microbial community phenotypes[40–42] (note that we will use the terms 'non-additive' and 'nonlinear' interchangeably throughout this paper). Specifically, one may extend this notion to define epistasis between environments, by comparing community properties observed on combinations of nutrient sets against those on individual nutrient sets.

Here, we determine how increasingly complex environmental compositions affect the growth yield and taxonomic structure of synthetic microbial communities. In addition to mapping the phenotypes of these communities along the axis of environmental complexity (the number of different carbon sources present in the medium), the design of our experiments allows us to quantify how communities are shaped by the combination of sets of carbon sources relative to their properties under each constituent set. By testing the effects of increasing numbers of up to 32 different carbon sources on over 280 synthetic microcosms, we examine how yield and diversity differ from expectations based on those in simpler environments. We further contextualize our results through the use of mathematical models, which reveal how these environment-phenotype relationships can be explained by a set of ecological rules for combining environments, with implications for the ecology of natural and engineered microbiomes.

## Results

**Assembly of communities in combinatorial environments**. We first designed microcosms with varying degrees of initial taxonomic and nutrient complexity (Fig. 1a-d). Based on experiments that assessed the resource utilization capabilities of various bacterial species (see Methods, Supplementary Fig. 1, Supplementary Fig. 2, Supplementary Table 1, Supplementary Table 2), we selected 13 organisms and 32 carbon sources intended to maximize taxonomic variability across environments. These organisms, which are not representative of any particular biome, were also chosen as they can be readily cultured individually and introduced into combinatorial environments in a controlled way. We generated increasingly complex combinations of our 32 carbon sources in a hierarchical manner, so that we could quantitatively compare the effects of higher-order combinations with those of simpler ones (Fig. 1d). Additionally, each environment contained the same amount of carbon irrespective of environmental complexity (Fig. 1b), enabling us to specifically assess the impact of increased resource heterogeneity. Our organisms were inoculated into these environments at equal amounts (see Methods, Supplementary Table 3, Supplementary Table 4), and the resultant cultures were grown and passaged into fresh media at rates informed by pilot experiments in order to maximize the chance of each having consumed the provided carbon sources and reached a stable composition (Supplementary Fig. 3, Supplementary Fig. 4b). In total, variants of this procedure were applied to generate 282 unique community-environment pairings (Supplementary Table 5).

**Community growth yield scaling with environmental complexity**. We initially asked whether and how the growth yield (defined as the difference between the maximum biomass at the end of growth and the initial biomass) of a community varies with increasing environmental complexity. To generate an expectation of this effect, we first employed a consumer resource modeling (CRM) framework. Consumer resource models, which predict community growth yields and composition based on species-specific resource utilization preferences (Fig. 1e), have previously been shown to recapitulate community dynamics in a variety of systems[43] and can be used to generate expectations at scales inaccessible to experiments. Moreover, they explicitly consider the dynamics and diversity of resources, making them especially well-suited to address questions relating to the ecological effects of environmental complexity. By using CRMs on a statistical ensemble of simulated communities (see Methods), we predicted that, on average, community yield would not change significantly with environmental complexity (Supplementary Fig. 5a). We then compared our simulation results to our in vitro 13-species community (referred to as com13, Supplementary Table 5), which, despite comprising a diverse set of organisms on heterogeneous carbon source combinations, closely matched this expectation (Fig. 2a). These models and experiments therefore suggested an additive relationship between community growth yield and environmental complexity. In other words, the overall yield for a complex community appeared to depend on average only on the total amount of carbon, and not on the number and identities of resources.

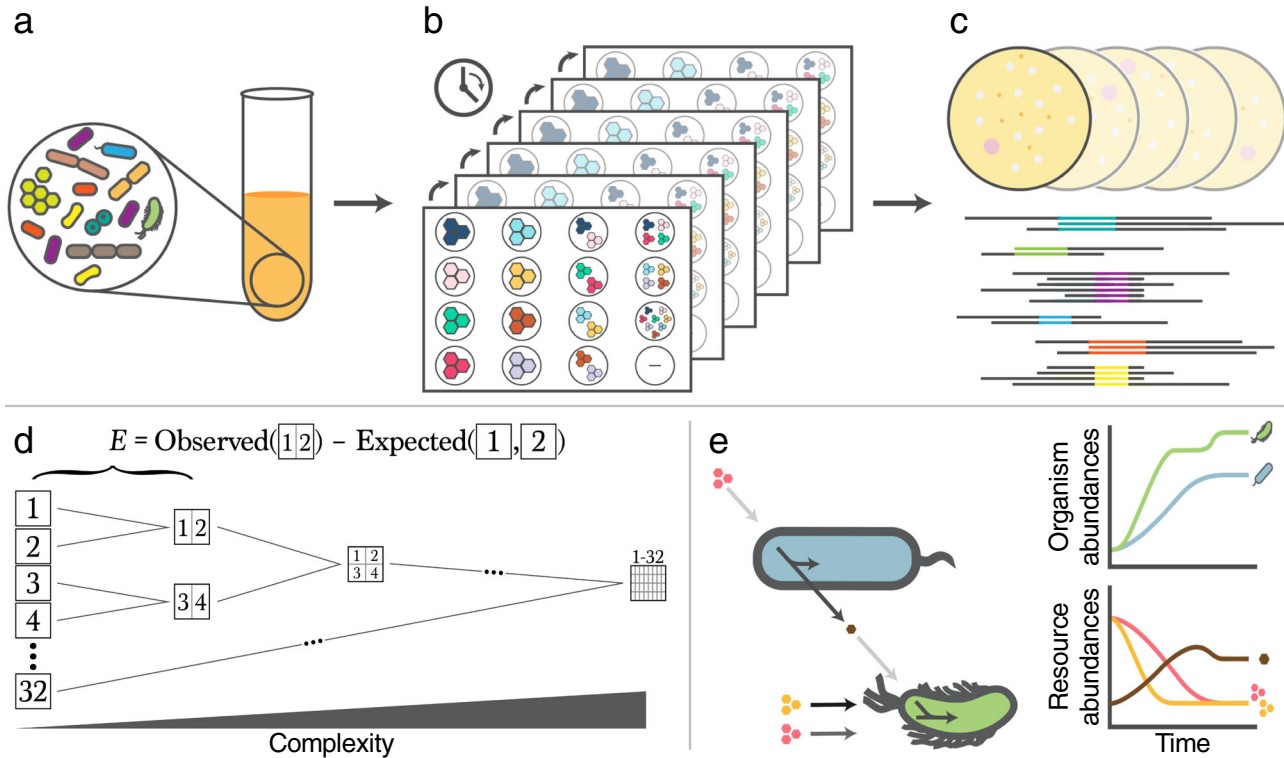

**Fig. 1 Experimental schematic for testing microbial community responses to environmental complexity. a** Communities were assembled by combining a defined number of organisms (Supplementary Fig. 1a, Supplementary Table 1) at equal ratios. **b** These mixed cultures were then inoculated into deep-well plates containing a minimal medium plus equimolar combinations of up to 32 carbon sources (Supplementary Fig. 1b, Supplementary Table 2, Supplementary Table 3, Supplementary Table 4). The communities were either grown in batch or diluted into fresh media over the course of several days (Supplementary Table 5). **c** Growth yields were then assessed using a spectrophotometer and composition was determined using either agar plating or 16S sequencing. **d** Experiments were designed such that community phenotypes in more complex environments could be directly compared to simpler environments containing the same carbon sources. Measurements of simpler environments were used to generate expectations of phenotypes in more complex compositions. **e** Schematic of consumer resource modeling framework. Resources (pink and yellow) are utilized by organisms to generate biomass and secreted byproducts (see Methods). In this example, the bottom green organism is able to utilize both resources more efficiently than the top blue one as denoted in the shades of the resource-organism arrows. The blue organism converts the pink resource into a brown metabolite, which can be utilized by the green organism for growth.

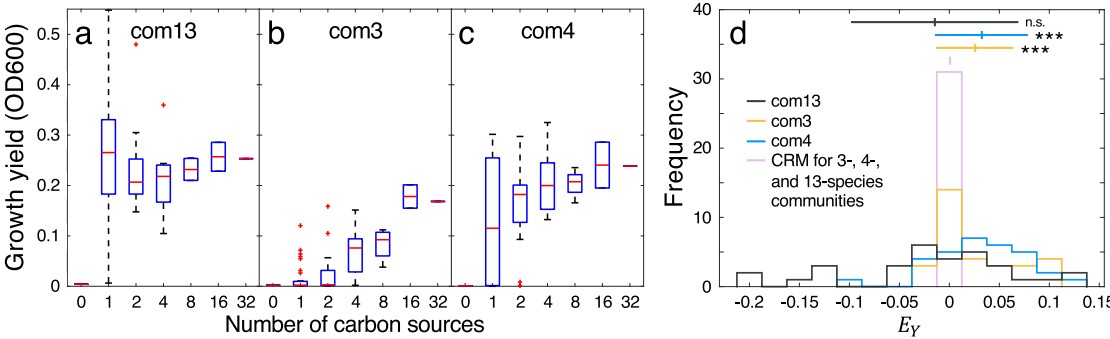

**Fig. 2 Changes in community growth yield in response to environmental complexity. a–c** Growth yields for com13 (**a**), com3 (**b**), and com4 (**c**) measured at the experimental endpoint (6 passages at 48-h frequencies, Supplementary Table 5). Here, the central mark indicates the median, the top and bottom box edges indicate the 25th and 75th percentiles, respectively, the whiskers extend to the most extreme points not considered outliers, and the red '+' symbols indicate outliers plotted individually. Sample sizes are outlined in Supplementary Table 3. **d** Distribution of yield epistasis $E_Y$ for all three communities and simulated communities predicted using a consumer resource model (CRM). $E_Y$ distributions for each community were compared against those of statistical ensembles of CRM-simulated communities containing the same number of initial organisms (i.e., 13, 3, or 4, Supplementary Fig. 5a, e, i). Here, histograms for the simulated communities completely overlap, so they are represented as a single peak. Upper bars denote mean and standard deviation. Significance levels are calculated between each community distribution and that of the CRM using a paired one-sided $t$-test, and are indicated by (***) $p < 0.001$ ($p = 0.84$ for com13, $3.6 \times 10^{-4}$ for com3, and $2.1 \times 10^{-4}$ for com4). Source data are provided as a Source Data file.

This effect can be further analyzed in a statistical way by using the concept of epistasis. In addition to confirming the aforementioned additivity, applying this concept to our communities allowed us to single out individual cases that significantly deviated from this general effect. To do this, we established a yield epistasis metric $E_Y$, which quantifies how much the yield $Y$ on a specific composite environment differs from the expectation based on its constituent environments (Fig. 1d). As such, our metric $E_Y$ is defined as the difference between the observed and expected yield on the combination of two environments $A$ and $B$, i.e.,

$$E_Y = Y(AB) - (Y(A) + Y(B))/2. \qquad (1)$$

An $E_Y$ value of zero would thus reflect the assumption that, since all environments contain the same total amount of carbon, the yield in a complex environment should be the same as the combination of yields on its corresponding simpler environments. Indeed, the distributions of yield epistasis scores $E_Y$ for our 13-species community (com13) and for the CRM simulations were centered at zero (Fig. 2d), confirming that our experiment and the corresponding model match our expectation of yield additivity. Using this metric, we then identified a number of notable deviations from $E_Y = 0$ (Supplementary Fig. 6). For example, the community cultured on the combination of D-glcNAc and D-galacturonate displayed an $E_Y$ value of 0.13 ($2\sigma$), indicating improved growth on this more complex composition than on the individual carbon sources. Conversely, the combination of D-glucose and D-sorbitol resulted in an $E_Y$ score of $-0.19$ ($3\sigma$), suggesting that the community might be displaying reduced efficiencies in using one carbon source in the presence of another, representing a type of 'resource interference' previously observed[17,44].

While our 13-species community matched the expectation that yields are additive on average, similar experiments on smaller (3- and 4-species) consortia, as well as on individual organisms, suggested that simpler microbial ensembles may not display this property. The yields observed in these experiments instead increased with environmental complexity (Fig. 2b, c, Supplementary Fig. 7a, Supplementary Fig. 8), an effect that was also reflected in a significant positive skewing of the distribution of $E_Y$ for many of these microcosms (Fig. 2d, Supplementary Fig. 8). This effect may stem from the resource utilization capabilities of the organisms involved. In a larger community, a broader representation of resource utilization capabilities may raise the chances that at least one organism will be able to consume the provided carbon at all levels of environmental complexity. In contrast, for smaller communities with more limited resource utilization capabilities, the chances of observing minimal or no growth would increase in simpler environments.

We directly tested this hypothesis by modifying our consumer resource model to account for the presence of organisms with more limited resource utilization capabilities (see Methods). In doing so, we noticed that yields did significantly increase in communities with reduced resource utilization potential (Supplementary Fig. 5). These simulations therefore suggest a metabolic mechanism underlying the trends observed in our smaller experimental consortia. Indeed, out of the 63 environments we tested, there were 33 and 15 in which our 3- and 4-species communities did not grow, respectively, compared to only 4 for our 13-species community (Supplementary Fig. 7a). Our 4-species community differed from our 3-species community only in the addition of one organism, *Pseudomonas aeruginosa*, whose broader and more efficient resource utilization capabilities (Supplementary Fig. 9) likely contributed to higher average growth in less complex conditions (Fig. 2b, c). Moreover, our communities displayed increasing average yields with initial

species richness (Supplementary Fig. 7b), in line with previous ecological observations[45–47].

Our simulations also revealed the role of organism relatedness in contributing to patterns of yield epistasis. As our in silico communities contained organisms with varying resource utilization capabilities, they also exhibited corresponding degrees of niche overlap (Supplementary Fig. 10). When we considered this ecological metric, we found that communities made up of more metabolically similar organisms exhibited more dampened increases in yield (Supplementary Fig. 5). In addition, parameterizing our models using the resource utilization capabilities measured for our communities (Supplementary Fig. 9) revealed their estimated degrees of niche overlap, while also recapitulating the relative magnitude of yield increases observed experimentally (Supplementary Fig. 5d, h, l). Taken together, our results suggest that an additive community growth yield response to environmental complexity depends on the number of organisms present, as well as the extent to which their individual metabolic capabilities overlap. These results, along with additional tests of community growth rates, thus point to an interplay between community-wide ecological effects and species-specific metabolic nonlinearities (Supplementary Note 1).

**Determinism and competition characterize community assembly.** Our analysis has so far focused on a single collective trait of microbial communities—the growth yield—but has not provided insight into how environmental complexity affects the balance between different organisms and their ensuing community structure. We thus used 16S amplicon sequencing to measure the endpoint taxonomic distributions of our 13-species community under increasingly complex environmental compositions (see Methods, com13, com13a, Supplementary Table 5). This analysis revealed considerable variation across different environments (Fig. 3a, Supplementary Fig. 11) and high degrees of consistency across replicates and experiments irrespective of environmental complexity (Supplementary Fig. 12, Supplementary Fig. 13), suggesting that the assembly patterns of these communities are largely deterministic based on environmental composition.

To more deeply analyze the contributions of defined carbon source sets to specific community structures, we applied a clustering analysis that yielded an environment-phenotype mapping spanning our entire dataset (Supplementary Fig. 14). This mapping demonstrated how distinct—and often unrelated—environments can nonintuitively result in similar taxonomic compositions, which reflect previously-identified family-level functional relationships in natural microbiomes (Supplementary Note 2, Supplementary Fig. 15)[24]. For example, we discovered a dichotomy between *Pseudomonas* and *Acinetobacter* organisms, which were the genera that dominated the communities in most (*Acinetobacter* in 18.0% and *Pseudomonas* in 76.7%) of our carbon source conditions. Our results showed that, while multiple organisms could generally persist in environments with single-carbon sources or with multiple different types of carbon sources (70.4% of environments resulted in the persistence of more than one organism), *Pseudomonas* organisms tended to dominate the communities in environments with more than one type of carbohydrate or organic acid. These outcomes thus underscore the importance of nutrient identity (and not just nutrient diversity) in determining the final structure of communities.

In addition to these patterns, we noticed how the overall species abundance distributions of our communities resembled those of natural microbiota in that they both generally contained a small number of high-abundance taxa and a long tail of low-abundance organisms[48–50] (Supplementary Fig. 16). Unlike natural ecosystems, however, knowing the composition of our initial inoculum

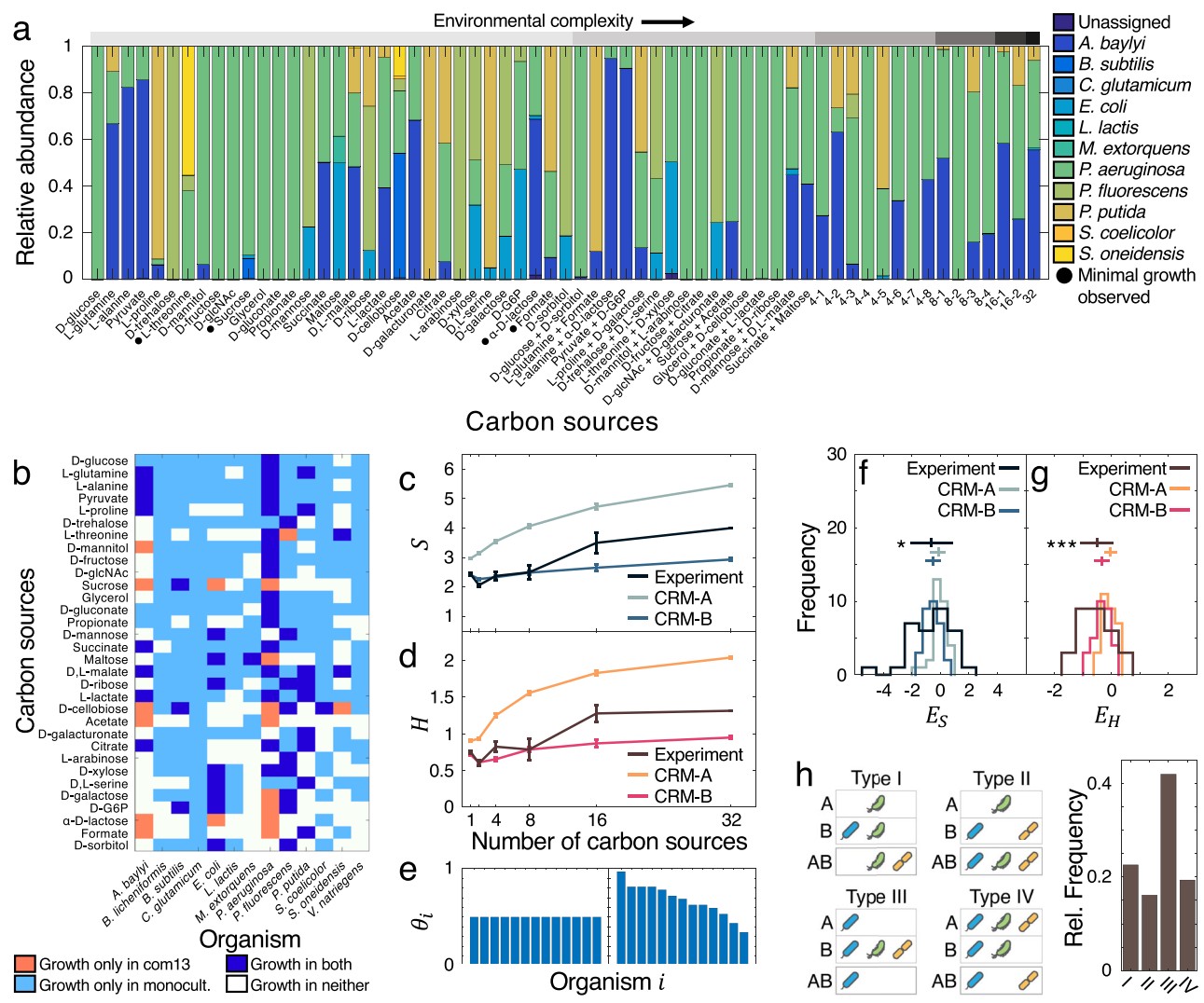

**Fig. 3 Endpoint taxonomic properties of 13-species community in up to 32 carbon sources (com13). a** Mean species relative abundances across three biological replicates. Environments with more than two carbon sources are abbreviated (e.g., condition 4–1 contains the carbon sources in the first two 2-carbon source conditions, etc.). For complete environmental compositions see Supplementary Table 3. Gray circles indicate community growth below OD$_{600}$ 0.05. Initial compositions and compositions across all replicates are found in Supplementary Fig. 13a. **b** Species-specific differences in growth between single-carbon source monoculture (Supplementary Fig. 9a) and single-carbon source community contexts. **c, d** Comparison of observed community species richness $S$ (**c**) and Shannon entropy $H$ (**d**) with phenotypes predicted by consumer resource models. The resource utilization capabilities of simulated organisms are either the same on average (CRM-A), or variable allowing for generalists and specialists (CRM-B). Data are represented as mean ± SEM. No significant increases in $S$ or $H$ were identified when comparing the single-carbon source cases to the 32-carbon source cases (one-tailed paired $t$-test $p = 0.107$ for $S$ and $0.180$ for $H$). **e** Representation of the fraction $\theta$ of resources usable by each organism $i$. Left: in CRM-A, each organism has the same low probability of consuming a given resource, resulting in low levels of niche overlap $\rho = 0.37 \pm 0.07$. Right: in CRM-B, this probability varies for each organism, determined by the fraction of carbon sources that were consumed by each organism in our monoculture experiments (Supplementary Fig. 9a). The composition of CRM-B communities resulted in higher degrees of niche overlap ($\rho = 0.50 \pm 0.09$). Organisms are displayed by decreasing $\theta$. **f, g** Distributions of species richness epistasis ($E_S$, **f**) and Shannon entropy epistasis ($E_H$, **g**) scores for experiment and CRM. Upper bars denote mean and standard deviation. Significance values are calculated against the distributions for CRM-A using a one-sided $t$-test and are indicated by (*) $p < 0.05$ and (***) $p < 0.001$ ($p = 0.034$ for $E_S$ and $1.26 \times 10^{-4}$ for $E_H$). **h** Schematic and prevalence of different epistasis types. Illustrations are representative examples. Type I: The environment AB results in the presence of an organism not observed in either environment A or B; Type II: AB results in the union of organisms from A and B; Type III: AB contains only the organisms from the lowest-diversity environment; Type IV: AB results in a more complex pattern of diversity loss. Source data are provided as a Source Data file.

allowed us to better understand the mechanisms that could be leading to these distributions. Specifically, we noticed that very few organisms out of the original 13 persisted in any single environment (Fig. 3a, Supplementary Fig. 11). This loss in diversity can be partially explained by competition, which was clearly visible in the single-carbon source environments. Here, many organisms did not persist in a community context despite being able to utilize a given carbon source in monoculture (Fig. 3b). This discrepancy was particularly striking in the D-glucose condition, in which, despite all but one organism being able to metabolize this carbon source (Supplementary Fig. 9), only *P. aeruginosa* remained. In addition, organisms that displayed a wide range of metabolic capabilities in monoculture, such as *C. glutamicum*, were not observed at all in the single-carbon source environments, suggesting that the structure of our communities was largely driven by competition. In support of this reasoning, we also found generally poor correlations between the growth yields reached by the organisms in monoculture and in com13 (Supplementary Fig. 17), indicating that monoculture resource utilization patterns are not necessarily predictive of how an organism will behave in a community. Despite the prevalence of interspecies competition, there were instances in which organisms unable to grow in monoculture on a given carbon source did survive in the same carbon sources in a community context (Fig. 3b). For example, the community grown on maltose contained *P. aeruginosa* despite its inability to utilize this carbon source (Supplementary Fig. 9a). A possible explanation is that *E. coli*, which was present in high abundance, secreted organic acids that sustained *P. aeruginosa* following the catabolism of maltose—a well-documented series of metabolic transformations[51,52].

**Negative epistasis dominates taxonomic diversity scaling.** The availability of hierarchical carbon source combinations (Fig. 1d) gave us the opportunity to extend our analysis of diversity to multi-carbon source conditions and ask whether, beyond anecdotal cases, general principles govern the scaling of diversity at increasing environmental complexity. We thus first used our 16S data to calculate the species richness $S$ and Shannon entropy $H$ values of our communities at each degree of environmental complexity—from 1 to 32 carbon sources. Our initial expectation that more carbon sources would create more niches and therefore lead to higher diversity in proportion to number of new molecules was challenged by the data, as neither diversity metric displayed a statistically significant increase as a function of environmental complexity (Fig. 3c, d). In fact, some single-carbon source conditions resulted in greater diversity than other more complex environments, suggesting that the number of resources is not a key determining factor of taxonomic diversity for these communities. Moreover, species co-occurrence patterns that we observed in single-carbon source environments were not preserved in more complex settings (Supplementary Fig. 18).

In order to more systematically assess the effects of combinations of environments on diversity, we defined epistasis metrics $E_S$ and $E_H$ for species richness and Shannon entropy, respectively. Like our epistasis scores for yield, these metrics quantify changes in taxonomic diversity based on expectations of these quantities on simpler environments:

$$E_S = S(AB) - \max(S(A), S(B));$$ (2)

$$E_H = H(AB) - \max(H(A), H(B)).$$ (3)

These scores use, as null expectations, lower bounds for $S$ and $H$ based on the intuition that a community on a more complex environment $AB$ should be at least as taxonomically diverse as that on $A$ or $B$. For example, if three species survived on

environment $A$, and two on environment $B$, $E_S = 0$ would represent the case in which the combination of environments $A$ and $B$ supported exactly three species. Alternatively, $E_S > 0$ (positive epistasis) would indicate that the joint environment $AB$ supported more than three organisms, while $E_S < 0$ (negative epistasis) would indicate that $AB$ supported fewer than three organisms. By computing these epistasis values for community compositions simulated using our consumer resource model, we found that the predicted distributions of $E_S$ and $E_H$ were both centered at zero (Fig. 3f, g; Supplementary Fig. 19c), confirming that our definition of epistasis constitutes a reasonable baseline to which we could compare our experimental results. In contrast to this basic expectation, but consistent with the low levels of diversity observed experimentally, the distributions of both scores for our in vitro 13-species communities were significantly skewed to the left ($E_S = -0.65 \pm 1.47$, $E_H = -0.50 \pm 0.58$; Fig. 3f, g; Supplementary Fig. 19d). In other words, our experiments revealed the pervasive presence of negative epistasis in how diversity behaves upon combining two sets of resources.

To better understand the causes underlying this phenomenon, we examined how the taxonomic compositions exhibited in individual environments translated to those in combinations of environments. We found that the taxonomic outcomes of combining two carbon source sets could be classified into four basic types (Fig. 3h, Supplementary Data 1). In about 20% of the cases displayed, the combined environment resulted in the appearance of one or more organisms that had not grown on the individual environments (Type I), suggesting the presence of beneficial interspecies interactions. However, the most common pattern emerging from our data was the dominance of the least diverse constituent environment (Type III, accounting for 40% of the cases). All instances of this dominance, which resembles complete buffering epistasis[39], were associated with strongly negative values of $E_S$, accounting in large part for the overall negative bias of the distribution.

The prevalence of this Type III pattern highlighted that even complex combinations of carbon sources often lead to the dominance of a single organism (Supplementary Data 1). We thus sought to determine whether this observation could be explained by explicitly considering the resource use capabilities of our organisms, as well as the degrees to which they intersect. To do this, we applied our consumer resource model to simulate two sets of 13-species communities: one based on the naïve assumption that all organisms consume the same limited number of resources on average (CRM-A), and another in which the proportion of resources usable by each organism was based on the number of carbon sources usable by each of our com13 organisms (Supplementary Fig. 9a), thereby reflecting the presence of metabolic generalists and specialists (CRM-B, see Methods, Fig. 3e). Communities in CRM-A also featured low degrees of niche overlap, while organisms in CRM-B had resource preferences that intersected to a greater degree (Supplementary Fig. 19c, d). We found that, while species richness and Shannon entropy were predicted to increase with environmental complexity in the CRM-A communities ($S$ reaching a maximum of ~6 coexisting species), they remained relatively flat in CRM-B communities (Fig. 3c, d) ($S$ reaching a maximum of ~3 coexisting species, a value and trajectory more reflective of our experimental observations). Predicted epistasis scores were also negatively skewed in CRM-B (Fig. 3f, g), suggesting that reduced taxonomic diversity in complex environments could be the outcome of competition in communities with uneven metabolic capabilities. A further generalization of our model to communities of different sizes revealed how increasing degrees of niche overlap are also negatively associated with taxonomic diversity (Supplementary Fig. 19), providing greater clarity on the low levels of diversity

observed in our 13-species communities as well as in smaller consortia assayed using agar plates (Supplementary Fig. 20a, b; Supplementary Note 3). Conversely, our model suggested that the presence of metabolic cross-feeding has the potential to slightly dampen losses in taxonomic diversity (Supplementary Fig. 19).

## Discussion

Deciphering how multispecies microbial communities grow on mixtures of resources remains highly challenging. Here, we showed how a hierarchical experimental design paired with extensive consumer resource modeling can be applied to address this question, revealing the role of resource specialization and niche overlap in determining the scaling of community properties with environmental complexity. Although our simplified experimental system is still far from the complexity of natural microbiomes[48,53], it captures properties that go beyond those observable in small artificial consortia. In particular, we identified a simple additive principle that explains how average growth yields can remain invariant with increasing environmental complexity—a consequence of all available resources being efficiently utilized given enough organisms with varied metabolic capabilities. Although one could expect this behavior to arise in communities well adapted to a specific environment, it is surprising that it also emerged in our synthetic consortia composed of organisms from different biomes grown on artificial combinations of carbon sources. Despite this additive relationship in some communities, our experiments and modeling showed how decreasing the degrees of community niche overlap could lead to non-additively increasing growth yields, reminiscent of observations of overyielding in various ecological studies[54–56]. However, while overyielding generally pertains to species mixtures displaying higher yields relative to monoculture, we describe a pattern by which these increases in yield are brought about by increasingly complex environments. To contextualize these observations, we drew from descriptions of nonlinearities in genetics and devised an 'epistatic' metric that allowed us to quantify our observed non-additive scaling of growth yield.

The versatility of the concept of epistasis allowed us to define similar metrics to quantitatively describe changes in taxonomic diversity. In contrast to our growth yield epistasis distributions that were either centered at zero or positively skewed, our distributions of diversity epistasis were centered on negative values. While the magnitude of negative epistasis was also highly dependent on organism resource specialization and niche overlap, our results raise the question of whether different distributions could be observed given an alternative formulation of our epistatic metrics. Indeed, the question of which mathematical definition best establishes a baseline for biological nonlinearities is a longstanding one in genetics[57,58], raising the prospect of new definitions as the basis for expanded quantitative evaluations of ecological nonlinearities. Irrespective of our formal definitions, however, our results showed how increased environmental complexity does not guarantee greater taxonomic diversity beyond that already possible on individual carbon sources[24]. This result underscores the dependence of biodiversity on an interplay of features, such an appropriate balance of generalists and specialists and the existence of evolved interdependencies[59–61]. Furthermore, it raises the prospect for systematic exploration of additional mechanisms that can impact the relationship between environmental complexity and community ecology. Of particular interest are experimental concerns such as the timescale and regime of medium dilutions[62], or metabolic ones such as the impacts of different molecular currencies (e.g., nitrogen or phosphorus[63–65]) and the ability of organisms to either sequentially or simultaneously utilize multiple resources[66,67]. Such

extensions would further clarify how communities respond to combinations of resources, facilitating the design of synthetic microbial ecosystems and improving multiscale models of communities adapted to complex environments, such as host-associated microbiomes and communities involved in biogeochemical cycles[68–72].

## Methods

**Selection and initial metabolic profiling of organisms.** In order to maximize the chance of obtaining communities with diverse taxonomic profiles from different environmental compositions, the organisms selected were drawn from a number of bacterial taxa known to employ varying metabolic strategies. In addition, given the growing relevance of synthetic microbial communities to industrial and biotechnological applications[73–76], we chose to employ bacterial species that have previously been used as model organisms and have well-characterized metabolic capabilities. This criterion, paired with the availability of flux-balance models associated with a majority of these organisms, allows us to explore the metabolic mechanisms observed in our various experimental conditions with higher confidence. These selection principles resulted in a set of 15 candidate bacterial organisms (*Acinetobacter baylyi*, *Bacillus licheniformis*, *Bacillus subtilis*, *Corynebacterium glutamicum*, *Escherichia coli*, *Lactococcus lactis*, *Methylobacterium extorquens*, *Pseudomonas aeruginosa*, *Pseudomonas fluorescens*, *Pseudomonas putida*, *Salmonella enterica*, *Streptomyces coelicolor*, *Shewanella oneidensis*, *Streptococcus thermophilus*, and *Vibrio natriegens*) spanning three bacterial phyla (Actinobacteria, Firmicutes, and Proteobacteria, Supplementary Table 1, Supplementary Fig. 1a).

A microtiter plate-based phenotypic assay was used to assess the metabolic capabilities of each of the 15 candidate organisms. Each organism, stored in glycerol at −80 °C, was initially grown in 3 mL of Miller's LB broth (Sigma–Aldrich, St. Louis, MO) for 18 h with shaking at 300 rpm at each organism's recommended culturing temperature (Supplementary Table 1). To maximize oxygenation of the cultures and prevent biofilm formation, culture tubes were angled at 45° during this initial growth phase. Candidate organism *Streptococcus thermophilus* was found to have produced too little biomass in this time period and was grown for an additional 8 h. Each culture was then separately washed three times by centrifuging at $6000 \times g$ for 2 min, removing the supernatant, suspending the pellet in 1 mL of M9 minimal medium with no carbon source, and vortexing or triturating to homogenize. The cultures were then diluted to $OD_{600}$ $0.5 \pm 0.1$ as read by a microplate reader (BioTek Instruments, Winooski, VT) and distributed into each well of three PM1 Phenotype MicroArray Plates (Biolog Inc., Hayward, CA) per organism at final $OD_{600}$ of $0.05 \pm 0.01$. The carbon sources in the PM1 plates (Supplementary Table 2, Supplementary Fig. 1b) were resuspended in 150 μl of M9 minimal media prepared from autoclaved M9 salts (BD, Franklin Lakes, NJ) and filter-sterilized $MgSO_4$ and $CaCl_2$ prior to inoculation. The cultures in each PM1 plate were incubated at each organism's recommended culturing temperature with shaking at 300 rpm for 48 h. After this growing period, the $OD_{600}$ of each culture was measured by a microplate reader to quantify growth. To account for evaporation in the outer wells of the plates, which could yield in inflated OD readings, three 'evaporation control' plates with no carbon source were inoculated with bacteria at a final $OD_{600}$ of 0.05 and incubated at 30 °C for 48 h. The averaged $OD_{600}$ readings of these plates were subtracted from the readings of the bacterial growth plates to correct for evaporation. A one-tailed $t$-test was performed using these corrected $OD_{600}$ values to determine significance of growth above the value of the negative controls ($p < 0.05$). These final growth yields for the 15 candidate organisms are reported in Supplementary Fig. 2a, and aggregated analyses of the growth profiles of the organisms are reported in Supplementary Fig. 2b-d.

After this initial metabolic profiling, *Streptococcus thermophilus* was not included in any of the subsequent experiments as it displayed too low of a growth rate in the initial overnight growth phase and grew very minimally (no more than $OD_{600}$ 0.2) in fewer than 20% of the carbon sources in the PM1 plate. After inclusion in an initial mixed-culture experiment (com14, Supplementary Table 5), *Salmonella enterica* was also removed from future experiments due to its high levels of growth on all but one of the PM1 plate carbon sources. Its exclusion, meant to prevent its complete dominance in the subsequent mixed-culture experiments, resulted in a final set of 13 bacterial organisms.

For experiments involving a subset of the 13 organisms, the organisms were chosen to ensure they could be differentiated via agar plating. In the 3-species community experiment involving *E. coli*, *M. extorquens*, and *S. oneidensis* in combinations of 5 carbon sources (com3a), the organisms were selected based on their easily differentiable colony morphologies (Supplementary Fig. 21). In the second 3- and 4-species community experiments (*B. subtilis*, *M. extorquens*, *P. aeruginosa*, and *S. oneidensis* (com3 and com4)), selection was informed by differentiable colony morphology and additional metabolic criteria based on generalist-specialist relationships. Absolute growth yield on the Biolog PM1 plates was also considered, with the goal of including both high- and low- yielding organisms. Therefore, *P. aeruginosa* (high-yield generalist), *B. subtilis* (low-yield specialist), *M. extorquens* (low-yield generalist), and *S. oneidensis* (high-yield specialist) were selected.

**Selection and combination of carbon sources.** The carbon sources used in all experiments were selected from the 95 carbon sources contained in the Biolog PM1 Phenotype MicroArray Plate. This plate contains a variety of molecule types such as mono- and disaccharides, sugar alcohols, organic acids, amino acids, and more complex polymers (Supplementary Table 2, Supplementary Fig. 1b). Using the metabolic profiling experiments for each individual organism as a basis (Supplementary Fig. 2a), different criteria were established to choose the carbon sources used in each experiment depending on the desired complexity of the environment. An overarching criterion was that each experiment contain at least one sugar, one organic acid, and one amino acid to increase the possibility of synergistic interactions between carbon sources and resource-use orthogonality between the organisms.

For the communities grown in 5 carbon sources (i.e., com3a and com13a in D-Glucose, pyruvate, D-glcNAc, L-proline, and L-threonine), the following criteria were applied: D-glucose was selected as it resulted in the highest yield of each of the individual organisms, pyruvate was an organic acid with relatively high yields, D-glcNAc was a more complex sugar that resulted in varying individual growth yields, and L-proline and L-threonine were amino acids that resulted in generally high and low individual species yields, respectively. Communities were grown in all combinations of these five carbon sources (5 conditions of 1 carbon source, 10 of 2, 10 of 3, 5 of 4, and 1 of 5) for a total of 31 unique environmental compositions (Supplementary Table 4).

The carbon sources for the 32-carbon source experiments were selected based on the following criteria, in decreasing order of importance: carbon sources in which generalists individually displayed low levels of growth but favored at least one specialist (3 carbon sources), carbon sources that resulted in high-variance in growth yields across organisms (5 carbon sources), and carbon sources that resulted in low-variance in growth yields across organisms (7 carbon sources). These criteria were meant to increase the probability of observing more taxonomically diverse communities. The remaining 21 carbon sources were selected based on the total organism-specific yields they conferred (Supplementary Fig. 2a), with higher-yielding carbon sources being prioritized. Communities were grown in selected combinations of these 32 carbon sources (32 conditions of 1 carbon source, 16 of 2, 8 of 4, 4 of 8, 2 of 16, and 1 of 32) for a total of 63 unique environmental compositions (Supplementary Table 3). The selected combinations were chosen based on the Biolog growth yields of the 13 organisms under each carbon source, with the lowest-yielding carbon source (D-sorbitol) being paired with the highest (D-glucose) followed by the second-lowest and second-highest, etc.

Growth media conditions were assembled by resuspending each carbon source in distilled water to stock concentrations of 1.25 mol C/L and filter sterilizing using 0.2 μm membrane filter units (Nalgene, Rochester, NY). Carbon source stock solutions were stored at 4 °C for no longer than 30 days. A liquid-handling system (Eppendorf, Hamburg, Germany) was used to distribute the individual carbon source stocks in the appropriate combinations in 96-well plates. These prepared carbon source stocks were then sterilized using filter plates (Pall Corporation, Port Washington, NY) via centrifugation at $1500 \times g$ for 2 min. These were then combined with M9 minimal medium (containing M9 salts, $MgSO_4$, $CaCl_2$, and no carbon) and filter-sterilized water to final working concentrations of 50 mM C in 96 deep-well plates (USA Scientific, Ocala, FL) for a total volume of 300 μl. This working concentration was selected such that all organisms would not grow beyond the linear range of $OD_{600}$ for biomass measurements.

**Culturing of microbial monocultures and communities.** After selecting the set of 32 carbon sources above, each of the 13 bacterial organisms was cultured independently in each individual carbon source prepared from stock solutions. Each organism was first inoculated from a glycerol stock stored at −80 °C into 3 mL of LB broth and incubated at 30 °C with shaking at 300 rpm for 18 h. The culture tubes were angled at 45° to prevent biofilm formation and to enable oxygenation of the cultures. The overnight cultures were then washed three times by centrifuging at $6000 \times g$ for 2 min, resuspending in M9 medium without carbon, and vortexing and triturating if necessary to homogenize. The individual cultures were then inoculated in biological triplicate into the prepared media plates at final concentrations in 300 μl of $OD_{600}$ 0.05 ± 0.01 as measured by a microplate reader (BioTek-Synergy HTX). Additionally, a control plate was assembled containing one well inoculated with each individual organism with no carbon source to assess the decay/evaporation of the initial inocula, and one uninoculated well with 50 mM C of D-glucose to control for contamination. These monocultures were grown at 30 °C with shaking at 200 rpm for 48 h, after which their growth yields were quantified by transferring 150 μl to clear 96-well plates (Corning, Corning, NY) and reading absorbance values ($OD_{600}$). Biomass quantities are reported as the difference between the raw $OD_{600}$ readings of each sample and the corresponding $OD_{600}$ value of the negative control wells. Outlying $OD_{600}$ readings were removed by calculating Z-scores $M$ for each individual measurement $x_i$ using the median absolute deviation (MAD):

$$M_i = \frac{0.6745(x_i - \tilde{x})}{\text{median}\{|x_i - \tilde{x}|\}} \qquad (4)$$

where $\tilde{x}$ is the median across three biological replicates and 0.6745 represents the upper quartile of the normal standard distribution, to which the MAD converges. If

the Z-score of an individual measurement exceeded 3.5, it was considered an outlier and removed. This process resulted in the elimination of 71 data points (out of 1248) across all organism monocultures. As with our Biolog phenotypic assay, a one-tailed t-test was performed using these corrected $OD_{600}$ values to determine significance of growth above the value of the negative controls ($p < 0.05$). These final growth phenotypes for the 13 organisms in stock solutions are reported in Supplementary Fig. 9 along with a comparison to their growth in the Biolog phenotypic assays.

For community experiments in combinatorial media, all communities were assembled using a bottom-up approach with each organism initially grown separately and diluted to the same starting concentrations before being combined. All individual organisms were prepared as described in the above paragraph, then combined at equal proportions and inoculated in biological triplicate into the prepared combinatorial media plates at final concentrations of $OD_{600}$ 0.05 ± 0.01 in 300 μl. Each community growth experiment additionally contained three control wells: one uninoculated well with 50 mM C of D-glucose to control for contamination, and two inoculated wells with no carbon source to assess the decay of the initial inocula. The communities were grown at 30 °C with shaking at 300 rpm for periods of 24 or 48 h before each passage. At each passaging step, the cultures were triturated 10 times to ensure the communities were homogenized and 10 μl were transferred to 290 μl of fresh media for the subsequent growth period. Yields of the cultures were quantified as described above, and processed using the aforementioned outlier removal procedure. This process resulted in the elimination of 8 data points (out of 192) for com3, 15 for com4, 10 for com13, and 17 (out of 768) across the four monocultures. A summary of the organisms, carbon sources, and culturing conditions for each community experiment is found in Supplementary Table 5.

Communities to be sequenced were centrifuged at $1500 \times g$ for 10 min and the supernatant was removed. Cell pellets were stored at −20 °C until DNA collection was performed using a 96-well genomic DNA purification kit (Invitrogen). To harvest the DNA, each cell pellet was resuspended in 180 μl lysis buffer containing 25 mM Tris-HCl, 2.5 mM EDTA, 1% Triton X-100, and 20 mg/ml Lysozyme (Sigma–Aldrich). The samples were mixed by vortexing and incubated at 37 °C for 30 min, after which 20 mg/ml of RNase A (Invitrogen, Carlsbad, CA) and 20 mg/ml of Proteinase K (Invitrogen) with PureLink Pro 96 Genomic Lysis/Binding Buffer (Invitrogen) were added. The samples were mixed by vortexing and centrifuged after each reagent was added. The samples were incubated at 55 °C for 30 min, after which 200 μl of 100% ethanol (Sigma–Aldrich) were added. DNA from the resulting lysates was purified using a vacuum manifold according to the purification kit protocol (Invitrogen). Purified DNA was normalized to 15 ng/μl using a NanoDrop 1000 spectrophotometer (Thermo Scientific, Waltham, MA). Library preparation was performed based on a paired-end approach developed by Preheim et al.[77], which targets the V4 region of the 16S rRNA gene with the primers U515F (5'-GTGCCAGCMGCCGCGGTAA) and E786R (5'-GGACTACH VGGGTWTCTAAT). Libraries were sequenced using an Illumina MiSeq platform at either the MIT BioMicroCenter, Cambridge, MA (com13a) or at QuintaraBio, Boston, MA (com13). QIIME2[78] was used to demultiplex raw files and produce FASTQ files for forward and reverse reads for each sample. DADA2[79] was used to infer sequence variants from each sample and a naïve Bayes classifier was used to assign taxonomic identities using a custom reference database with a 95% confidence cutoff. Reads not able to be assigned to a genus in our database were marked as 'Unassigned' and comprised 0.19% and 0.01% of all reads across all samples for com13 and com13a, respectively.

Communities to be assayed by agar plating were diluted by a factor of $10^4$ and spread on LB agar plates using autoclaved glass beads. Plates were prepared by autoclaving and distributing 18 mL of LB agar (Sigma–Aldrich) into petri dishes using a peristaltic pump (TriTech, Los Angeles, CA). Inoculated plates were incubated at 30 °C and imaged after 72 h using a flatbed scanner for colony counting. Colony counts for com3 and com4 were adjusted based on a standard dilution of the community members at equal concentrations measured by OD.

Significance between growth yields under differing environments was determined using a one-sided two-sample t-test with significance cutoffs of 0.05, 0.01, and 0.001. Species richness ($S$) is defined as the number of different organisms detected in a particular environment. Shannon entropy ($H$) is defined as follows:

$$H = -\sum_i p_i \log_2 p_i \qquad (5)$$

where $p_i$ is the relative abundance of organism $i$ in a sample.

For hierarchical clustering analysis of communities, Spearman correlation coefficients were computed either for pairs of environments or pairs of organisms based on normalized vectors of species abundances. Hierarchical clustering was performed on the correlation coefficients using the 'clustergram' function in MATLAB, which calculated distances between clusters using the UPGMA method based on Euclidean distance.

**Computation of epistasis scores.** To quantify the non-additivity in how taxonomic diversity and balance could change in incrementally more complex environments, we first established definitions of expected values of species richness $S$ and Shannon entropy $H$ based on their values in lower-complexity conditions. Let a combined set of carbon sources $AB$ be defined as the union of carbon source sets $A$ and $B$. For carbon source sets $A$, $B$, and $AB$, the vectors of relative species

abundances in each set are defined as $V_A$, $V_B$, and $V_{AB}$, respectively. The species richness values $S$ for each set are therefore simply the number of positive species abundance values in each vector. Based on the organisms that survived in sets $A$ and $B$, we establish the naïve assumption that at least as many organisms as survived in either environment will also survive in set $AB$. We therefore define $S_{expected, AB}$ as max $(S_A, S_B)$. This value is then compared to the experimentally observed species richness value $S_{AB}$ using our epistasis score for species richness $E_{S, AB}$:

$$E_{S,AB} = S_{AB} - S_{expected,AB}. \qquad (6)$$

We use a similar expectation to calculate Shannon entropy epistasis $E_H$. Here, we first calculate the observed $H$ for carbon source set $AB$ as $H_{AB} = -\sum V_{AB} \log_2 V_{AB}$, and the expected $H$ for carbon source set $AB$ based on $A$ and $B$ as $H_{expected, AB} = \max (H_A, H_B)$. The epistasis score for Shannon entropy $E_{H, AB}$ is therefore:

$$E_{H,AB} = H_{AB} - H_{expected,AB}. \qquad (7)$$

**Consumer resource modeling.** We employed a dynamical modeling framework to simulate the yields of arbitrary communities in increasingly complex environments, as well as the relative abundances of com4. The model builds upon Robert MacArthur's consumer resource model[80–82] and a subsequent modification by Marsland et al.[43], which simulates the abundances of organisms over time as a function of resource availability, metabolic preferences, and exchange of secreted metabolites.

We define the individual species abundances as $N_i$ for $i = 1,...,S$, and the resource abundances as $R_\alpha$ for $\alpha = 1,...,M$. The key variable in calculating the abundances $N_i$ is a stoichiometric resource utilization matrix $C_{i\alpha}$, which defines the uptake rate per unit concentration of resource $\alpha$ by organism $i$ (Fig. 1e). To calculate the growth of each organism on each resource, we multiply this matrix by a Monod consumption term $R_\alpha/(k_{i,a} + R_\alpha)$ that simulates concentration-dependent resource depletion. Each consumed resource type $\alpha$ with abundance $R_\alpha$ is therefore consumed by organism $i$ at a rate $C_{i\alpha}R_\alpha/(k_{i,a} + R_\alpha)$. These resources $\alpha$ are then transformed into other resources $\beta$ by the organisms via a species-specific normalized stoichiometric matrix $D_{\alpha\beta i}$. A fraction $l$ of the resultant metabolic flux is returned to the environment to be made available to other organisms, while the rest is used for growth. In addition to these resource consumption terms, the species abundances are also defined by (i) a species-specific conversion factor from energy uptake to growth rate $g_i$, (ii) a scaling term representing the energy content of each resource $w_\alpha$, and (iii) a quantity representing the minimal energy uptake required for maintenance of each species $m_i$. These terms are further defined in Supplementary Table 7. Taken together, the species abundances $N_i$ over time are defined by:

$$\frac{dN_i}{dt} = g_i N_i \left[ \sum_\alpha w_\alpha (1 - l_\alpha) C_{i\alpha} \frac{R_\alpha}{k_{i,a} + R_\alpha} - m_i \right]. \qquad (8)$$

The initial resource abundances $R_{\alpha,0}$ and the initial organism abundances $N_{i,0}$ are first defined; then each resource is consumed in a manner dependent on the matrix $C_{i\alpha}$, and converted to other resources based on the stoichiometric matrix $D_{\alpha\beta i}$:

$$\frac{dR_\alpha}{dt} = \sum_i C_{i\alpha} N_i \frac{R_\alpha}{k_{i,a} + R_\alpha} + \sum_{i,\beta} D_{\alpha\beta i} \frac{w_\beta}{w_\alpha} l_\beta C_{i\beta} N_i \frac{R_\beta}{k_{i,\beta} + R_\beta}. \qquad (9)$$

We selected the parameters for our equations based on experimental observations and quantities obtained from the literature (Supplementary Table 7). Our model contains no terms for resource replenishment or culture dilution, as all species were diluted every 48 h at a proportion defined by our experiments (10 μl of culture passaged into a total of 300 μl of fresh uninoculated media). Kinetic growth curves for com14 (Supplementary Table 5, Supplementary Fig. 22a) were used to estimate the orders of magnitude for the remaining parameters, based on the community reaching an average of approximately $2.4 \times 10^8$ CFU/mL (OD$_{600}$ 0.3) within 20 h in 50 mM C of D-glucose. The conversion factor from energy uptake to growth rate $g_i$, as well as the energy content of each resource $w_\alpha$, were set to 1 and $1 \times 10^8$, respectively, in order to approximate this magnitude of growth. The Monod resource consumption half-velocity constant was set to $1 \times 10^4$ g/mL for all resources in order to approximate the experimentally observed growth timeframe. Lastly, the minimum energy requirements for all organisms $m_i$ were informed by the community yields at steady state and the leakage fraction $l_\alpha$ was set to 0.8 based on community simulations in Marsland et al.[43]. These quantities are summarized in Supplementary Table 7.

We used our model to simulate the growth of communities containing $S = 13$, $S = 3$, and $S = 4$ arbitrary organisms. Here, values in $C$ could only be nonzero if a randomly defined probability $P^{i\alpha}_{util}$ of an organism $i$ being able to utilize a particular resource $\alpha$ was below a given threshold value $\theta_i$. This threshold, representative of the fraction of resources usable by each organism, ranges from 0 to 1 and can be decreased to make an organism more of a resource specialist. Each community was simulated 50 times in each environment, so that the resource preference matrices $C$ and the resource utilization probabilities $P^{i\alpha}_{util}$ could be randomly repopulated. This process allowed us to more effectively sample the large space of possible resource utilization matrices and obtain a clearer indication of how mean community yields changed in response to increasing environmental complexity. The environments

were generated from a set of $M = 32$ arbitrary resources, which were combined in a scheme similar to that of com3, com4, and com13: 32 conditions with one resource, 16 conditions with two resources, and so on, up until one condition with all 32 resources.

We initially simulated our communities with a $\theta_i$ of 1 for all organisms $i$, as an initial null model for assessing how growth yields could vary with increasing environmental complexity (Supplementary Fig. 5a, e, i). These communities, which had high degrees of niche overlap (Supplementary Fig. 10) and distributions of yield epistasis $E_Y$ centered at zero, were used as a baseline to quantify the yield epistasis distributions of our in vitro communities (Fig. 2d). We next carried out simulations with decreasing values of $\theta_i$ (and correspondingly decreasing degrees of niche overlap), which recapitulated the increases in yield we observed experimentally for com3 and com4 (Supplementary Fig. 5f-g, j-k). We then carried out simulations in which we defined each value $\theta_i$ as the proportion of nutrients able to be consumed by each organism in our monoculture assays, $\theta_{mc}$ (Supplementary Fig. 9a, Supplementary Fig. 5d, h, l). Two of these simulation results (that of a 13-species community with $\theta_i = 0.5$ for all $i$, and of a 13-species community with $\theta = \theta_{mc}$) were used as CRM-A and CRM-B, respectively, in our main analysis of taxonomic diversity with generalists and specialists.

To test the effects of metabolic exchange, our CRM simulations also contained between 1 and 10 unique secreted metabolites, which could also be consumed by organisms according to randomly defined preferences in $C$. For a more realistic representation of metabolic conversion, these byproducts were matched with primary resources in conversion matrix $D$, which was randomly populated according to a transition probability of 0.25, meaning that a given metabolic byproduct had a 25% chance of being converted from a given primary resource. This matrix was normalized across each primary resource to ensure conservation of mass. Our results for yield, species richness, and Shannon entropy are presented as the average across all quantities of metabolic byproducts. The initial species and resource abundances were set to $6 \times 10^6$ CFU/mL and 1.5 g/mL, respectively, to approximate the initial OD$_{600}$ of 0.05 and the initial resource concentration of 50 mM C of glucose used in our experiments. All communities were simulated over the course of 288 h with dilutions every 48 h (based on com3, com4, and com13 culturing timescale) with a timestep of 0.01 h.

Our calculation for the degree of niche overlap $\rho$ of our simulated communities was based on a previous formulation of the metric designed for consumer resource models[83]. This metric, bounded between 0 and 1, is 0 if the organisms do not compete for resources and is 1 if the organisms' resource utilization profiles completely overlap. It is defined using the mean $\mu$ and variance $\sigma^2$ of the community resource utilization matrix $C$ as:

$$\rho = \frac{\mu_C^2}{\mu_C^2 + \sigma_C^2}. \qquad (10)$$

**Reporting summary.** Further information on research design is available in the Nature Research Reporting Summary linked to this article.

## Data availability

The authors declare that the data supporting the findings of this study are available within the paper, its Supplementary Information files, and a permanent GitHub repository at github.com/segrelab/EnvironmentalComplexity[84]. Categorized community taxonomic outcomes on combinations of carbon sources are provided in Supplementary Data File 1. Source data are provided with this paper.

## Code availability

Code for analyzing data and for running consumer resource simulations is available at github.com/segrelab/EnvironmentalComplexity.

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

## Acknowledgements

We thank Christopher P. Mancuso, Jennifer M. Bhatnagar, Sylvie Estrela, and Ilija Dukovski for experimental advice and technical assistance. We are also grateful to Sean P. Mullen for access to the robotic liquid-handling system and to James E. Fifer for relevant experimental training. Additionally, we thank Pankaj Mehta, Einat Segev, and Daniel Sher, as well as past and present members of the Segrè group for helpful discussions on the research and the manuscript, particularly David B. Bernstein, Joshua E. Goldford, Robert Marsland III, Demetrius DiMucci, Elena Forchielli, and Devlin Moyer. A.R.P. is supported by a Howard Hughes Medical Institute Gilliam Fellowship and a National Academies of Sciences, Engineering, and Medicine Ford Foundation Pre-doctoral Fellowship. We gratefully acknowledge support from the U.S. Department of Energy, Office of Science, Office of Biological & Environmental Research through the Microbial Community Analysis and Functional Evaluation in Soils SFA Program (m-CAFEs) under contract number DE-AC02-05CH11231 to Lawrence Berkeley National Laboratory, as well as the National Institutes of Health (grants NIDCR R01DE024468, NIGMS R01GM121950, NIA UH2AG064704), the National Science Foundation (grants 1457695 and NSFOCE-BSF 1635070), the Human Frontiers Science Program (grant RGP0020/2016), and the Boston University Interdisciplinary Biomedical Research Office.

## Author contributions

A.R.P. and D.S. designed the research. A.R.P. and M.L.O. designed the experiments. A.R.P. performed experiments, collected data, wrote data analysis code, developed the models, and ran and analyzed simulations. All authors wrote, read, and approved the final manuscript.

## Competing interests

The authors declare no competing interests.
