## [Peer Review File · Nature Communications]

Reviewers' Comments:

Reviewer #1:

Remarks to the Author:

Overview: The paper by Pacheo and Segre examine how diversity and functioning (i.e., biomass yield) are influenced by the number of substrates supplied to consortia of bacteria. The authors find that yield (i.e. maximum biomass) changes additively. In other words, biomass yield does not change for a given microbial community when the number of carbon sources is manipulated. However, the diversity of the microbial community changes with the number of substrates. Specifically, it appears that diversity (S and H) increases with increasing number of substrates (Fig. 3 c, d), but perhaps less so than expected based on models (Fig. 3 f, g).

1) Before reading the paper, the title made me think that this study would be about something very different. "Environmental nonlinearity of microbial ecosystems" made me envision some sort of nonlinear function, perhaps how biomass yield changes as a function of substrate number. The paper doesn't generate this sort of figure (but maybe Fig. 3c-e?). Instead, the study documents non-additive relationships that arise when different bacteria are exposed to combinations of different numbers of substrates. It's questionable whether it's appropriate to frame the study in terms of "ecosystems" as the experiments are conducted at extremely small spatial and temporal scales (microtiter plates).

2) The authors draw analogy to non-additive relationships by discussing epistasis. I don't have a problem with this and it should help evolutionary biologist understand the motivation. However, this is not an evolution experiment. The non-additive interactions arise via ecological processes involving different species. So, this could be confusing to some readers. The authors may want to consider focusing (in addition) to the many ecological experiments that have been done, which are similar to the current study in many ways. Throughout the early 2000s (and still today), ecologists have been interested in how variables like yield, stability, nutrient cycling changes with different number of species. A lot of work has gone into the nuanced issues of how to design these experiments (randomly vs. non-randomly constructing communities from regional assemblages), but also how to interpret the findings. What is being referred to here as positive epistasis, is overyielding (and underyielding, for the opposite) among ecologists who think about these patterns. I would recommend that the authors look into the biodiversity ecosystem functioning (BEF) literature, as it seems to be very relevant to the questions in the current study.

3) Substrate choices. In the methods, the authors provide some justification for deciding how substrates were chosen. It seems that the much of the study was based on Biolog plates. These plates are convenient because they can be easily ordered, but it's unclear how this choice affects the inferences that are made in the end. Starting on line 357, the authors describe how they grouped substrates into different classes (e.g., sugar, organic acids, amino acids). Then, there are additional criteria based on generalist and specialist growth responses presumably under monoculture conditions. It's unclear to me how these decisions regarding substrate combination might have affected the results, but it definitely does not seem like it was done randomly. The consumer resource models make it clear that the authors are thinking about stoichiometric balance. I'm not sure what currencies are being considered though. For example, is this C, N, P, Fe, etc.? Or other macromolecular characteristics? I suspect other properties of the substrates might also be important, for example size, bond complexity, or energy content (i.e., delta G). Given that the authors are working with a tractable and well characterized set of substrates, it seems like these would be interesting and generalizable properties to consider.

4) Thirteen strains of bacteria end up being the focus of this study. The authors describe how two of these strains (*Streptococcus* and *Salmonella*) were excluded. The remaining strains, which we are told belong to the Actinobacteria, Firmicutes, and Proteobacteria, were retained because of their growth characteristics and because of their relevance to synthetic biology and industrial applications. The names are finally listed in Supplementary Figure 2. Three of the strains belong to the genus

Pseudomonas and many of the strains appear to be well-behaved fast-growing strains that are commonly used in model systems. If this is a fair assessment, then I think it is reasonable to ask about the generality of the findings. One thing that I would recommend is that the authors consider how phylogenetic relatedness affects the patterns. For example, is positive epistasis more likely or unlikely if a consortium is made up of highly related taxa? If we assume that more closely related strains are more likely to have similar metabolic capacities, then one might expect that the strains would have overlapping niches and lower E_y values. A formal test would involve checking to see if there is phylogenetic signal. If there is, there are ways to correct for this.

5) Flux balance models for four microbial taxa are described starting on line 492. One can imagine that this approach could be useful for understanding how organisms behave in consortia since the modeling could potentially help predict cross-feeding and inhibition. It is unclear, however, how the flux balance modeling of only four (instead of 13) species is being used. Furthermore, it is not obvious how the flux balance modeling is incorporated into the major findings. Are these somehow used to inform the consumer resource models?

6) Consumer resource modeling is used to generate null expectations for the 13 taxa. It appears that these models predict lower epistasis for yield (Fig 2d) and diversity (Fig. 3.f.g) compared to observations. Does this reflect something special going on in with the biology or is there something not accurately represented in the consumer resource models? There are a lot of terms and parameters. Some of these appear to have been obtained from the literature (lines 557). Are these values sufficient for describing the interactions of the focal organisms in this study? How is dilution rate in the model (e.g., chemostat?) align with the batch-culture conditions of the experiment? In the end, the major conclusions of the study don't seem to rest very strongly on the simulations. My interpretation is that the empirical data deviate from model predictions. That could be because a) something interesting (synergistic) is occurring, or b) the models are missing important features. Perhaps more could be done to support the first interpretation.

Minor comments:

1) The authors are inconsistent when referring to the microbial substrates. Throughout the ms, they refer to them as "nutrients", "carbon sources", "environmental composition", "environmental molecules", "substrates", etc. It would be clearer if they chose one term. Also, "yield" is not well defined or consistently used (e.g. Fig. 2).

2) "16s" should be capital S.

3) Figure 2: does average yield increase with the richness of the community? Panels are organized strangely ($S = 13, 3, 4$), but it appears that such a trend might exist, which would be consistent with ecological expectations.

Jay Lennon
Indiana University

Reviewer #2:

Remarks to the Author:

In this paper, Pacheco and Segre investigate how increasing the number of carbon sources shapes the yield and diversity of synthetic bacterial consortia initiated with up to 13 species. By keeping the total amount of carbon constant across environments, they find that the community yield generally remains constant regardless of the number of resources. Community diversity in mixed nutrient environments, however, is lower than expected by a purely additive model, and is most often similar to the least diverse environment. By coupling these experiments with consumer-resource modeling, they suggest

that this low diversity may be due competition between generalists and specialists.

This paper investigates a topic of great interest and relevance to microbiome research. The manuscript is clearly written, the methods are well-described, and some interesting results are presented. Below are some comments and suggestions that I think could help improve the manuscript.

1) Figure 2d shows the distributions of yield epistasis for 3 synthetic communities (com3, com4 and com13) as well as the distribution for one simulated community (CRM-com13). The model for com13 matches quite well the experiment, but how do the model and experiment compare for smaller communities? Showing the model-predicted yields for com3 and com4 would be helpful here.

2) Figure 3b. In the legend, it reads "Species-specific differences in growth between monoculture (Biolog assay, Supplementary Figure 2a) and single-carbon source community contexts." Does this mean that the single-species (monoculture) and multi-species communities were grown in different plates/ assay conditions? If so, how does growth in the Biolog plates compare to growth in plates where the carbon sources were prepared from stock solutions (as described lines 383-386)? Please clarify.

3) P5, lines 150-151. It reads "We thus used 16s amplicon sequencing to measure the endpoint taxonomic distributions of two 13-species communities (com13, com13a, Supplementary Table 5) under increasingly complex environments (see Methods)." This sentence seems to suggest there are two different 13-species communities both assembled under increasingly complex environments. But looking at Table S5, com13 and com13a consist of the same 13 species and what differs is the number of nutrients (13 nutrients for com13 and 5 nutrients for com13a). Please clarify in the text.

4) Figure 3A and Figure S9. What does unassigned mean? Does it correspond to a single ESV or multiple ESVs? Even if the species-level is not assigned, is the genus-level assigned? Please clarify.

5) P5, lines 163-167. It reads "... Pseudomonas and Acinetobacter organisms, which were the genera that dominated the communities in most of our nutrient conditions. Our results showed that, while organisms could generally coexist in environments with single nutrients or with multiple different types of nutrients, Pseudomonas organisms dominated the communities in environments with more than one type of carbohydrate or organic acid." What % does "in most" and "generally" correspond to?

6) P13, line 483. Please define the term S_AB in the text.

7) P14, line 548, it should read Supplementary Table 9 not Supplementary Table 7.

8) P16, lines 614-629. I generally find the section describing the generalist/specialist simulations not very clear. Lines 618-620, it reads "an organism was classed as a generalist if it was able to grow on more than 90% of the nutrients (e.g. *P. aeruginosa*), and a specialist if it was able to grow fewer than 50% of the nutrients (e.g. *B. subtilis*)." How are species able to grow on less than 90% but more than 50% classified? Also, it reads "For our first 13-species community" (line 621) and "In our second 13-species community (line 622)". I am assuming these correspond to CRM-A and CRM-B. Please clarify. Finally, what is the rationale for using different criteria to define the nutrient consumption probability in com4 vs com13?

Reviewer #3:

Remarks to the Author:

This is a fantastic study that investigates how yield and diversity scale with nutrient complexity. The

question is timely and helps to establish general principles governing microbial community properties. The main outcome is non-intuitive, and thus has the potential to be paradigm shifting. The study design is appropriate and the main conclusions are well supported by the results. Moreover, the manuscript is very well written, being both clear and concise. I only have a few points I would encourage the authors to consider.

1) One of the most fascinating outcomes is that diversity can decrease as nutrient complexity increases. This is attributed to generalists that excel at mixed substrate utilization. While the authors explanation for this observation is convincing, it seems to conflict with the idea of a tradeoff between metabolic specialization and generalization (e.g, specialists achieve higher rates and/or substrate-specific yields than generalists, but can only use a subset of the available substrates). In this study, the authors assembled communities consisting of rather distantly related organisms, which could result in trait differences that mask a generalization-specialization tradeoff. If the authors were to assemble communities consisting of closely related strains with few trait differences other than different breadths of metabolic capabilities (e.g. by genetic engineering different KO mutants), would the authors expect the same outcome?

2) Regarding generalists; I can delineate two types of generalists. A) Generalists that consume a wide range of substrates simultaneously. B) Generalists that consume a wide range of substrates sequentially, and are therefore effectively specialists at any give point in time. If I understand correctly, type A generalists were modeled in this study. However, type B generalists are pervasive in nature. How would considering type B generalists impact the interpretation of the data?

3) While I believe that investigating yield and diversity already represents a major advance in our understanding of microbial communities, I continuously wondered why the authors did not also investigate community growth rates. I assume these data were obtained during operation of the plate readers, and I am therefore curious about how growth rates related to substrate complexity. I do NOT think including such data into this study is necessary, but I wonder if the authors would at least comment on this.

MINOR COMMENTS

Line 20: I am not a huge fan of the term 'molecular complexity'. The term 'molecular' is ambiguous in this context. I would prefer the term 'nutrient complexity'.

Line 165: I would be careful with the term "co-existence". An unambiguous demonstration of co-existence typically requires a reciprocal invasion experiment.

Reviewer #4:

Remarks to the Author:

The authors present a systematic study of microbial resource competitions where sets of specific species are grown on sets of specific carbon sources. The study is admirable for the care taken to allow for comparisons between outcomes, e.g. by providing a fixed amount of carbon for each competition and combining species and carbon sources in ways that allow for inference of epistatic interactions. There are several results worthy of note. A clear result is that small numbers of species (with no "generalist" species) often fail to achieve maximal final biomass. The authors attribute this to a mismatch between the provided nutrients and the repertoire of nutrients consumable by the species present. While this outcome is not particularly surprising, it has not been definitively demonstrated before to my knowledge. A more interesting observation is the prevalence of negative epistasis for final diversity of species upon mixing two sets of resources. Overall, I found the work to be a solid contribution to address an important question, namely how the variety of nutrients influences microbial diversity. That said, my enthusiasm for the work would be substantially higher if the authors

could provide more mechanistic insight into the origin(s) of the observed negative epistasis. I believe this could be accomplished by improvements to the modeling without requiring additional experiments, as discussed below.

Major issues:

1. Modeling. In my view, the current version of the model misses a critical aspect of the experiments - specifically the serial dilution protocol. Resource competition in a chemostat and in serial dilution can be quite different because in the former resources are constant, whereas there is a time course of nutrient availability in the latter. I believe that some of the results, including the negative epistasis among nutrients, may only be understandable by explicitly considering the nutrient time course. For example, a recent study by Erez and coworkers (eLife 2020) identified an early bird effect in which species that grew best early in serial dilution outcompeted late growers due to the early increase of the former's population. This would seem to be consistent with the observed negative epistasis: if a species has an early growth advantage on either of the nutrient sets, it would also be an early bird on the mixed set, also leading to reduced diversity. The authors should reformulate their model to better match the serial dilution protocol of the experiments. This would be a major improvement to the manuscript, and would provide valuable insight into the intriguing observation of negative epistasis. Optimistically, it might even be possible to provide mechanistic insight into the double power law behavior seen in Fig. S14A.

2. Discussion. The discussion more or less just repeats the highlights of the Results, and barely puts the work in any broader context. Additional insights from modeling would help here, as would consideration of relevant previous studies. Ecologists have been studying the relationship between ecosystem yield/function and diversity for a long time and it is still an area of intense study. It would be helpful to discuss how the current study fits into this spectrum, e.g. Langenheder et al. PLOS ONE 2010 as just one example. This comes back again to the minimal consideration of mechanism in the current work. How could different results regarding richness vs. nutrient diversity reflect different mechanisms at work?

Minor issues:

3. This phrase in the abstract was particularly opaque until I had read the paper: "...rules that govern how communities respond nonlinearly to the coupling of different nutrient sets". I'm sure this can be rephrased to provide more insight to someone reading through the abstract.

4. Some additional value would be added to Fig. 3a by clustering similar outcomes, at least for the single nutrient cases.

5. For the analysis of negative epistasis between nutrient sets in Figs. 3f,g, rather than only taking as a baseline prediction the larger parent diversity, it would be worth also showing the results taking as a baseline the smaller parent diversity.

6. On lines 223 and 224, the same inequality is written for both positive and negative epistasis, the second inequality should be flipped.

7. It's unclear what the authors have tested with the t-tests in Fig. 2d. They state that the distributions are centered at zero, but then show their means are statistically different from a zero centered distribution. This should be clarified.

8. There is a repeated typo in the Monod consumption formula: it should be a plus sign in the denominator, so $R/(k + R)$.

Reviewer #1 (Remarks to the Author):

Overview: The paper by Pacheo and Segre examine how diversity and functioning (i.e., biomass yield) are influenced by the number of substrates supplied to consortia of bacteria. The authors find that yield (i.e. maximum biomass) changes additively. In other words, biomass yield does not change for a given microbial community when the number of carbon sources is manipulated. However, the diversity of the microbial community changes with the number of substrates. Specifically, it appears that diversity (S and H) increases with increasing number of substrates (Fig. 3 c, d), but perhaps less so than expected based on models (Fig. 3 f, g).

Dr. Lennon, thank you for reviewing our article and for your helpful comments. We are glad you found our study interesting and have addressed your remarks as follows:

1. Before reading the paper, the title made me think that this study would be about something very different. "Environmental nonlinearity of microbial ecosystems" made me envision some sort of nonlinear function, perhaps how biomass yield changes as a function of substrate number. The paper doesn't generate this sort of figure (but maybe Fig. 3c-e?). Instead, the study documents non-additive relationships that arise when different bacteria are exposed to combinations of different numbers of substrates. It's questionable whether it's appropriate to frame the study in terms of "ecosystems" as the experiments are conducted at extremely small spatial and temporal scales (microtiter plates).

We thank you for bringing this to our attention, and agree that the current title may not best express the message and scope of our study. In order to more sharply address our focus on communities (as opposed to natural ecosystems), as well as our hierarchical experimental design, we have changed our title to "Non-additive microbial community responses to environmental complexity."

2. The authors draw analogy to non-additive relationships by discussing epistasis. I don't have a problem with this and it should help evolutionary biologist understand the motivation. However, this is not an evolution experiment. The non-additive interactions arise via ecological processes involving different species. So, this could

be confusing to some readers. The authors may want to consider focusing (in addition) to the many ecological experiments that have been done, which are similar to the current study in many ways. Throughout the early 2000s (and still today), ecologists have been interested in how variables like yield, stability, nutrient cycling changes with different number of species. A lot of work has gone into the nuanced issues of how to design these experiments (randomly vs. non-randomly constructing communities from regional assemblages), but also how to interpret the findings. What is being referred to here as positive epistasis, is overyielding (and underyielding, for the opposite) among ecologists who think about these patterns. I would recommend that the authors look into the biodiversity ecosystem functioning (BEF) literature, as it seems to be very relevant to the questions in the current study.

Thank you for this suggestion and for the opportunity to better contextualize our work in the broader microbial ecology literature. We have added a section to the Discussion comparing yield epistasis to the concept of over/underyielding in BEF (Line 517 in the document with tracked changes), with the caveat that since we are extending the concept of epistasis to taxonomic diversity, we found this language to be more appropriate for our study. In doing so, we have compared our findings with those drawn from key BEF papers (PMID: 24904563, PMID: 31997566, PMID: 24904563).

3. Substrate choices. In the methods, the authors provide some justification for deciding how substrates were chosen. It seems that the much of the study was based on Biolog plates. These plates are convenient because they can be easily ordered, but it's unclear how this choice affects the inferences that are made in the end. Starting on line 357, the authors describe how they grouped substrates into different classes (e.g., sugar, organic acids, amino acids). Then, there are additional criteria based on generalist and specialist growth responses presumably under monoculture conditions. It's unclear to me how these decisions regarding substrate combination might have affected the results, but it definitely does not seem like it was done randomly. The consumer resource models make it clear that the authors are thinking about stoichiometric balance. I'm not sure what currencies are being considered though. For example, is this C, N, P, Fe, etc.? Or other macromolecular characteristics? I suspect other properties of the substrates might also be important, for example size,

bond complexity, or energy content (i.e., delta G). Given that the authors are working with a tractable and well characterized set of substrates, it seems like these would be interesting and generalizable properties to consider.

The reviewer is correct that we employed a systematic process to select the carbon sources for our community experiments. Given the limitations of our experimental scale, we aimed to select carbon sources that would maximize the chance of supporting biodiverse communities in our environments. We chose this approach (as opposed to a random selection of carbon sources), as we wanted to identify a realistic upper bound for final community taxonomic diversity. To do this, we first cultured all of our bacteria on Biolog plates as they represent an accessible means to test organisms' metabolic potential on a reasonably wide breadth of substrates. We used this data to inform our selection of the 32 carbon sources, chosen using the following criteria in decreasing order of importance:

1. Carbon sources in which generalists individually displayed low levels of growth but favored at least one specialist
2. Carbon sources that resulted in high-variance in growth yields across organisms
3. Carbon sources that resulted in low-variance in growth yields across organisms
4. Carbon sources that conferred high yields to individual organisms

We describe this rationale in greater detail in the Methods (Line 703). Though we believe our selection resulted in a diverse spread of community compositions, we agree that explicitly considering additional molecular currencies (e.g. N, P, S, Fe) can also shed light on the patterns we observed. Though we are not aware of existing work on how community function and diversity scale with increasing amounts of nitrogen or phosphorus for example, the complex ways organisms utilize and balance different currencies are likely to have a major effect on emergent community properties (PMID: 28508070, PMID: 24739236).

Although our experimental and modeling design can be readily extended to identifying these contributions, we believe that a more complete assessment of the

impact of different currencies would be best achieved via additional targeted studies. Nonetheless, we carried out a basic analysis of how different carbon source types (carbohydrates, organic acids, and amino acids) impacted the overall taxonomic compositions of our 13-species community (Supplementary Figure 15). In doing so, we identified a slight increase in taxonomic balance in communities that contained an amino acid, though it is unclear whether this is due to the presence of additional nitrogen. We have added a line to the Discussion (Line 598) to acknowledge the relevance of extending our method to these additional currencies.

4. Thirteen strains of bacteria end up being the focus of this study. The authors describe how two of these strains (*Streptococcus* and *Salmonella*) were excluded. The remaining strains, which we are told belong to the Actinobacteria, Firmicutes, and Proteobacteria, were retained because of their growth characteristics and because of their relevance to synthetic biology and industrial applications. The names are finally listed in Supplementary Figure 2. Three of the strains belong to the genus *Pseudomonas* and many of the strains appear to be well-behaved fast-growing strains that are commonly used in model systems. If this is a fair assessment, then I think it is reasonable to ask about the generality of the findings. One thing that I would recommend is that the authors consider how phylogenetic relatedness affects the patterns. For example, is positive epistasis more likely or unlikely if a consortium is made up of highly related taxa? If we assume that more closely related strains are more likely to have similar metabolic capacities, then one might expect that the strains would have overlapping niches and lower E_y values. A formal test would involve checking to see if there is phylogenetic signal. If there is, there are ways to correct for this.

Thank you for this suggestion. Firstly, we have added a list of all of our organisms to the first paragraph of the Methods section in order to make our choice of bacteria more clear (Line 615). The reviewer's assessment of our community selection process is also correct: as we aimed to focus exclusively on the ecological effects of environmental complexity, we chose to keep our initial species compositions as balanced as possible. This was achieved by first culturing all of our organisms individually in rich medium, and then combining them in equal proportions for our experiments. This protocol thus required relatively fast-growing strains whose growth

capabilities were suited to our culturing conditions, which resulted in the final list of 13 organisms tested.

We nonetheless agree that the generality of our findings could be limited given our relatively narrow breadth of community compositions. Although natural microbial ecosystems can contain thousands of different species (PMID: 31822687, PMID: 20203603), it has been shown that a small subset of taxa often have an outsize effect on community structure and function (PMID: 32200744, PMID: 29789680). Importantly, the *ex situ* culturing of communities in liquid minimal medium is known to enrich for a much smaller number of organisms (PMID: 30072533, PMID: 31787942), suggesting that our study would have encountered similarly limited levels of species richness had we cultured natural samples. Nonetheless, examining communities from the perspective of functional niches can shed light on species coexistence, even in environments with a single limiting nutrient (PMID: 30072533).

With the reviewer's comments and these considerations in mind, we examined how our observations of yield and diversity epistasis could depend on species relatedness. To do this, we significantly expanded the scope of our consumer resource modeling to simulate communities with varying degrees of ecological niche overlap. While niche overlap is not the only available metric for quantifying species relatedness, it reflects the functional differences that exist between organisms and aligns well with the metabolic focus of our study. Moreover, it can be readily defined for consumer resource models (PMID: 30636966, see Methods Line 1211).

We assembled *in silico* communities of varying sizes (3, 4, and 13 initial organisms) with different degrees of niche overlap ρ , carrying out 415,800 unique simulations in total (63 environmental conditions \times (0,10) available secreted metabolites \times 50 random samplings \times 3 community sizes \times 4 degrees of niche overlap, see Methods Line 1109). As the reviewer expected, our simulation showed lower E_Y values in communities with greater degrees of niche overlap (Supplementary Figure 5), aligning with our previous observation that broader resource utilization capabilities lead to more complete resource consumption in simpler environments. Parameterizing our consumer resource models using experimentally-measured resource utilization preferences revealed the estimated degrees of niche overlap of

our *in vitro* communities, while also recapitulating the relative magnitude of yield increases observed experimentally (Supplementary Figure 5d, h, l). In addition to shedding light on our observations of yield epistasis, our modeling also showed how decreasing niche overlap enabled greater species coexistence (Supplementary Figure 19), supporting our experimental observations of the prevalence of interspecies competition (Figure 3b). We have highlighted these results in the Results (Line 263 and Line 480), and have also integrated niche overlap into the Introduction (Line 45) and Discussion (Line 586).

5. Flux balance models for four microbial taxa are described starting on line 492. One can imagine that this approach could be useful for understanding how organisms behave in consortia since the modeling could potentially help predict cross-feeding and inhibition. It is unclear, however, how the flux balance modeling of only four (instead of 13) species is being used. Furthermore, it is not obvious how the flux balance modeling is incorporated into the major findings. Are these somehow used to inform the consumer resource models?

Flux-balance modeling was used to investigate the increases in growth yield observed for com3 (*B. subtilis*, *M. extorquens*, and *S. oneidensis*) and com4 (com3 + *P. aeruginosa*). Specifically, we sought to understand whether the number of unique metabolites secreted by these organisms could contribute to a greater nutrient pool (and therefore the observed increases in yield) in more complex environments. As curated genome-scale models exist for these organisms, we used flux-balance analysis (FBA) to obtain the number and identity of the metabolites predicted to be secreted across all of our carbon source combinations. Our FBA modeling showed that the number of unique secreted metabolites plateaus at relatively low degrees of environmental complexity (Supplementary Figure 25), perhaps forming a basis for the 'diminishing returns' in terms of growth yield we observed in more complex environments for these communities (Figure 2b, c, Supplementary Text).

Moreover, knowing the identities and secretion rates of these metabolites allowed us to parameterize consumer resource models (CRMs) of these specific communities. For these simulations, our FBA results directly fed into our formulation of the *D* matrix, which defines rates of carbon source-specific conversion. These CRMs were able to

recapitulate the number of organisms observed experimentally, as well as the dominance of *P. aeruginosa* in most conditions of com4 (Supplementary Figure 27). However, as *M. extorquens* was observed to grow on a large number of carbon sources in monoculture, our CRM overpredicted its prevalence in a community context in contradiction to our experimental results.

As our use of flux-balance models was limited to these applications in this study, we have moved our description of the procedure used for FBA to a Supplementary Methods section for greater clarity. Nonetheless, we would like to highlight the novelty of directly parameterizing dynamical models of interspecies competition with mechanistic metabolic predictions, and will continue to develop this hybrid approach in future work.

6. Consumer resource modeling is used to generate null expectations for the 13 taxa. It appears that these models predict lower epistasis for yield (Fig 2d) and diversity (Fig. 3.f.g) compared to observations. Does this reflect something special going on in with the biology or is there something not accurately represented in the consumer resource models? There are a lot of terms and parameters. Some of these appear to have been obtained from the literature (lines 557). Are these values sufficient for describing the interactions of the focal organisms in this study? How is dilution rate in the model (e.g., chemostat?) align with the batch-culture conditions of the experiment? In the end, the major conclusions of the study don't seem to rest very strongly on the simulations. My interpretation is that the empirical data deviate from model predictions. That could be because a) something interesting (synergistic) is occurring, or b) the models are missing important features. Perhaps more could be done to support the first interpretation.

We thank the reviewer for this point, and agree that while our modeling did employ average dilution and resource replenishment rates that were in line with our experiment, they did not specifically simulate the serial dilution protocol we used. We therefore redesigned our consumer resource model to explicitly recreate our serial dilution process (see Methods, Line 1086). We used this updated protocol to re-run all CRM simulations, and have regenerated the relevant figures that depend on these modeling results. As a representative example, we direct the reviewer to

Supplementary Figure 26, which shows *in silico* analogs of com3 and com4 growing on multiple resources. This figure shows the effects of dilutions every 48 hours on organism and resource abundances.

Integrating serial dilutions at 48-hour time points also allowed us to better understand the distributions of yield epistasis E_Y that we observed experimentally. Specifically, when our model was parameterized with the resource utilization profiles of our organisms, it was able to recapitulate the increases in yield displayed by com3 and com4 (Supplementary Figure 5h, l). We had previously been unable to recapitulate these yield increases with our chemostat CRM, which suggests that the specific experimental dilution regime is an important factor in determining the steady-state yield of the community (Line 598).

These results support our hypothesis that a greater diversity of resources leads to a higher probability that specialist organisms will be able to utilize any resource, leading to higher average yields (Line 263). Though our model also predicted increases in yield for our 13-species communities (Supplementary Figure 5d), the magnitude of these increases was very small compared to those of our 3- and 4-species communities. These differences further support an interplay between community species richness and resource utilization capabilities in determining growth yields and potential nonlinearities.

In addition to allowing us to better contextualize the distribution of yield epistasis, incorporating serial dilutions into our models also provided valuable insight on our observations of taxonomic diversity. In this case, however, our models predicted a stronger negative skewing of species richness and Shannon entropy epistasis when incorporating generalists and specialists (CRM-B, Figure 3c, d, f, g). These resulted in an underprediction of the experimentally-observed diversity metrics compared to our previous chemostat model. Nonetheless, our serial dilution version of CRM-B still correctly predicted a negatively-skewed distribution of E_S and E_H compared to CRM-A, reinforcing our observation that uneven resource use profiles contribute to lower-than-expected degrees of taxonomic diversity.

We thank the reviewer again for this suggestion, and have also updated our scripts available on Github (github.com/segrelab/EnvironmentalComplexity) to include the serial dilution modeling protocol.

Minor comments:

7. The authors are inconsistent when referring to the microbial substrates. Throughout the ms, they refer to them as "nutrients", "carbon sources", "environmental composition", "environmental molecules", "substrates", etc. It would be clearer if they chose one term. Also, "yield" is not well defined or consistently used (e.g. Fig. 2).

Thank you for pointing this out. In order to more consistently describe our work, we now employ the following terminology throughout the manuscript:

- a. "Carbon source" when referring to our experimental process and results.
- b. "Resource" for language related to organism metabolic capabilities and our consumer resource models.
- c. "Environment" when referring to the pool of available resources in our experiments and models.
- d. "Nutrient" limited to references of relevant literature or of growth-supporting molecules in a general sense (including carbon sources).
- e. "Substrate" no longer used.

We have also more clearly defined our use of the term "yield" (Line 170) and have made all mentions of it in the text and figures consistent to this definition.

8. "16s" should be capital S.

This has been corrected throughout the manuscript.

9. Figure 2: does average yield increase with the richness of the community? Panels are organized strangely ($S = 13, 3, 4$), but it appears that such a trend might exist, which would be consistent with ecological expectations.

We did observe an increasing yield with increasing community richness, which we have now formally quantified in Supplementary Figure 7c. We have also highlighted this relationship in the text (Line 259), and made references to key studies that contextualize this phenomenon (PMID: 30301905, PMID: 26010833, PMID: 17594423). Thank you for pointing this out and for your suggestion.

Jay Lennon
Indiana University

Reviewer #2 (Remarks to the Author):

In this paper, Pacheco and Segre investigate how increasing the number of carbon sources shapes the yield and diversity of synthetic bacterial consortia initiated with up to 13 species. By keeping the total amount of carbon constant across environments, they find that the community yield generally remains constant regardless of the number of resources. Community diversity in mixed nutrient environments, however, is lower than expected by a purely additive model, and is most often similar to the least diverse environment. By coupling these experiments with consumer-resource modeling, they suggest that this low diversity may be due competition between generalists and specialists.

This paper investigates a topic of great interest and relevance to microbiome research. The manuscript is clearly written, the methods are well-described, and some interesting results are presented. Below are some comments and suggestions that I think could help improve the manuscript.

We thank the reviewer for their positive evaluation of our paper and for their helpful comments.

1. Figure 2d shows the distributions of yield epistasis for 3 synthetic communities (com3, com4 and com13) as well as the distribution for one simulated community (CRM-com13). The model for com13 matches quite well the experiment, but how do the model and experiment compare for smaller communities? Showing the model-predicted yields for com3 and com4 would be helpful here.

We have modified the calculations behind Figure 2d and now compare our experimentally-observed E_Y distributions to those of simulated communities with the corresponding number of organisms. These predictions resulted from a substantial expansion of our consumer resource modeling, which are shown in greater detail in Supplementary Figure 5 and are also referenced in the caption of Figure 2. As the E_Y distributions drawn from our null model were centered at zero for all three community sizes, they appear as a single peak in the histogram of Figure 2d. We have modified the figure legend and caption to clarify this point.

2. Figure 3b. In the legend, it reads "Species-specific differences in growth between monoculture (Biolog assay, Supplementary Figure 2a) and single-carbon source community contexts." Does this mean that the single-species (monoculture) and multi-species communities were grown in different plates/ assay conditions? If so, how does growth in the Biolog plates compare to growth in plates where the carbon sources were prepared from stock solutions (as described lines 383-386)? Please clarify.

We thank the reviewer for this excellent point. We chose to use our initial Biolog phenotypic assay to make this comparison as (1) the Biolog carbon sources were resuspended in the same base M9 minimal medium as our stock carbon sources, and (2) Figure 3b only deals with a comparison of binary growth capabilities (as opposed to growth rates or yields). We believe that this allowed us to quantify the role and prevalence of interspecies competition in a community context.

However, the question the reviewer raised prompted us to specifically consider the differences in phenotype between Biolog plates and stock solutions, since to our knowledge no such direct comparison exists for the number of strains and carbon sources we studied. We therefore carried out an additional set of experiments where we cultured each individual strain in each of the 32 carbon sources prepared from stock solutions. These experiments were performed using the same culturing conditions as those for our multispecies experiments (Line 751 in the document with tracked changes), enabling a more direct interrogation of the role of interspecies competition in our communities. The results of these experiments are reported in Supplementary Figure 9a, which shows the diversity of metabolic capabilities of our organisms. These results were then used to update Figure 3b, which reconfirmed the absence of many of our organisms in a community context despite their ability to metabolize various carbon sources. These species-specific resource use capabilities were also used to re-parameterize our consumer resource models, whose predictions were in close agreement with our previous simulations (Figure 3c-g).

We nonetheless noticed varying degrees of inconsistency between monoculture growth in Biolog plates vs. in stock solutions (Supplementary Figure 9b). Overall, we found an agreement of 73.6% between culturing methods, suggesting the possibility

of additional unreported metabolites in the Biolog plates or inconsistent carbon source concentrations. We contacted Biolog Inc. to inquire as to the exact chemical composition of the PM1 Phenotype MicroArray plate, but were told that it was proprietary. We therefore hope that our direct comparison of organism growth in Biolog carbon sources vs. in those prepared from pure stocks, while outside the scope of our study, can serve as an additional resource for groups who may wish to perform similar experiments in the future.

3. P5, lines 150-151. It reads "We thus used 16s amplicon sequencing to measure the endpoint taxonomic distributions of two 13-species communities (com13, com13a, Supplementary Table 5) under increasingly complex environments (see Methods)." This sentence seems to suggest there are two different 13-species communities both assembled under increasingly complex environments. But looking at Table S5, com13 and com13a consist of the same 13 species and what differs is the number of nutrients (13 nutrients for com13 and 5 nutrients for com13a). Please clarify in the text.

Thank you for pointing this out. We have clarified the wording in the text (Line 290).

4. Figure 3A and Figure S9. What does unassigned mean? Does it correspond to a single ESV or multiple ESVs? Even if the species-level is not assigned, is the genus-level assigned? Please clarify.

A specific ESV is marked as 'Unassigned' if our naïve Bayes classifier was not able to match it to any of the genera contained in our custom database. Such reads amounted to 0.19% and 0.01% of all reads across all samples for com13 and com13a, respectively. We have clarified our usage of the term in the classification process in the Methods section (Line 865).

5. P5, lines 163-167. It reads "... Pseudomonas and Acinetobacter organisms, which were the genera that dominated the communities in most of our nutrient conditions. Our results showed that, while organisms could generally coexist in environments with single nutrients or with multiple different types of nutrients, Pseudomonas organisms dominated the communities in environments with more than one type of

carbohydrate or organic acid." What % does "in most" and "generally" correspond to?

We have clarified these quantities in the corresponding section in the text (Line 307).

6. P13, line 483. Please define the term S_{AB} in the text.

Thank you for pointing out this omission, we have defined the term more clearly (Line 933).

7. P14, line 548, it should read Supplementary Table 9 not Supplementary Table 7.

The typo has been corrected.

8. P16, lines 614-629. I generally find the section describing the generalist/specialist simulations not very clear. Lines 618-620, it reads "an organism was classed as a generalist if it was able to grow on more than 90% of the nutrients (e.g. *P. aeruginosa*), and a specialist if it was able to grow fewer than 50% of the nutrients (e.g. *B. subtilis*)." How are species able to grow on less than 90% but more than 50% classified? Also, it reads "For our first 13-species community" (line 621) and "In our second 13-species community (line 622)". I am assuming these correspond to CRM-A and CRM-B. Please clarify. Finally, what is the rationale for using different criteria to define the nutrient consumption probability in com4 vs com13?

Thank you for pointing out this lack of clarity. In response to a different reviewer's comment, we have expanded our modeling to account for a wider breadth of community sizes and degrees of niche overlap (Line 1100). As a result, the specific generalist-specialist cutoffs mentioned by the reviewer are no longer used or mentioned. The reviewer is also correct in assuming we were referring to CRM-A and CRM-B in lines 621-622 of the original manuscript. We have re-worded this passage to clarify which communities are being referenced in the revised version (Line 1114). Lastly, the nutrient consumption probabilities for all simulated organisms are now defined by our new monoculture growth experiment (Supplementary Figure 9).

Reviewer #3 (Remarks to the Author):

This is a fantastic study that investigates how yield and diversity scale with nutrient complexity. The question is timely and helps to establish general principles governing microbial community properties. The main outcome is non-intuitive, and thus has the potential to be paradigm shifting. The study design is appropriate and the main conclusions are well supported by the results. Moreover, the manuscript is very well written, being both clear and concise. I only have a few points I would encourage the authors to consider.

We are glad that the reviewer found our work interesting, and are grateful for their positive assessment of our manuscript and their helpful comments.

1. One of the most fascinating outcomes is that diversity can decrease as nutrient complexity increases. This is attributed to generalists that excel at mixed substrate utilization. While the authors explanation for this observation is convincing, it seems to conflict with the idea of a tradeoff between metabolic specialization and generalization (e.g, specialists achieve higher rates and/or substrate-specific yields than generalists, but can only use a subset of the available substrates). In this study, the authors assembled communities consisting of rather distantly related organisms, which could result in trait differences that mask a generalization-specialization tradeoff. If the authors were to assemble communities consisting of closely related strains with few trait differences other than different breadths of metabolic capabilities (e.g. by genetic engineering different KO mutants), would the authors expect the same outcome?

We thank the reviewer for this point, and agree that the generality of our findings could be limited by our relatively narrow breadth of community compositions. To address this limitation, we significantly expanded the scope of our consumer resource modeling to simulate communities with more closely- or distantly-related organisms by examining their degrees of niche overlap. While this is not the only available metric for quantifying species relatedness, niche overlap is reflective of the functional differences that exist between organisms and aligns well with the metabolic focus of our study. Moreover, it can be readily defined for consumer

resource models (PMID: 30636966, see Methods Line 1100 in the document with tracked changes).

We assembled *in silico* communities of varying sizes (3, 4, and 13 initial organisms) with different degrees of niche overlap ρ , carrying out 415,800 unique simulations in total (63 environmental conditions \times (0,10) available secreted metabolites \times 50 random samplings \times 3 community sizes \times 4 degrees of niche overlap, see Methods Line 1114). Our simulations showed that communities with greater degrees of niche overlap had more dampened yield epistasis trajectories (Supplementary Figure 5), in agreement with our previous observation that broader resource utilization capabilities lead to more complete resource consumption in simpler environments. We also parameterized our models using experimentally-measured resource utilization preferences, which revealed the estimated degrees of niche overlap in our *in vitro* communities and recapitulated the relative magnitude of yield increases observed experimentally (Supplementary Figure 5d, h, l).

Our expanded modeling also showed how decreasing niche overlap enabled greater species coexistence (Supplementary Figure 19), supporting our experimental observations of the impact of interspecies competition on taxonomic diversity (Figure 3b). We have highlighted these results in the Results (Line 263 and Line 480), and have also integrated niche overlap into the Introduction (Line 45) and Discussion (Line 586).

2. Regarding generalists; I can delineate two types of generalists. A) Generalists that consume a wide range of substrates simultaneously. B) Generalists that consume a wide range of substrates sequentially, and are therefore effectively specialists at any give point in time. If I understand correctly, type A generalists were modeled in this study. However, type B generalists are pervasive in nature. How would considering type B generalists impact the interpretation of the data?

This is a great point, and we agree that a mechanistic analysis of different modes of resource consumption can further clarify the ecological patterns we observed. Indeed, the contexts under which organisms may employ either diauxie or co-utilization are varied (PMID: 30894528, PMID: 32561713), and are highly pertinent to our results as

they have been described for several organisms used in our study (PMID: 24799698, PMID: 32184246). Members of our group are currently working to explicitly integrate these modes of consumption into consumer resource models, which we hope will further shed light on how the distribution of generalists and specialists impact the scaling of taxonomic diversity. Though we feel that thoroughly answering these questions will require additional studies outside the scope of this work, we have added a mention of this future direction in the Discussion (Line 598).

3. While I believe that investigating yield and diversity already represents a major advance in our understanding of microbial communities, I continuously wondered why the authors did not also investigate community growth rates. I assume these data were obtained during operation of the plate readers, and I am therefore curious about how growth rates related to substrate complexity. I do NOT think including such data into this study is necessary, but I wonder if the authors would at least comment on this.

We agree with the reviewer that understanding how community growth rates may vary with environmental complexity is invaluable to the study of microbial ecology. In fact, despite primarily focusing on endpoint growth yield and taxonomic diversity, we did collect growth rate data for one of our 13-species communities (com13a, Supplementary Figure 3b). This time course was primarily used as a pilot experiment to help determine the appropriate timescale for our main dilution experiments, but also provided some insight into how growth rates could be affected by increased environmental complexity (Supplementary Figure 23a). Namely, our community displayed decreasing generation times with increasing environmental complexity, which is consistent with previous experimental observations (PMID: 23963223; PMID: 30894528). While our discussion of this experiment was limited to the Supplementary Text, we have added a clearer reference to it in the Results section (Line 274).

We nonetheless believe that a broader evaluation of this effect is necessary, as this experiment was limited to one single community on only five carbon sources. While the reviewer is correct that growth rate data can be readily obtained using the same plate reader-based methods we employed, continuous monitoring of cell cultures was out of reach for the majority of experiments carried out in our study. This is

because:

1. In order to mitigate the effects of evaporation during our growth periods, we used an experimental volume of 300 μl in 96-deep well plates, which exceeds the capacity of shallow well plates able to be read in the spectrophotometer. Though our reported measurements of OD600 were performed at every dilution by transferring 150 μl of culture into shallow well plates, our cultures lacked the volume to allow us to perform these transfers at short time intervals during exponential growth. Such continuous sampling from our cultures would have also increased the risk of contamination.
2. Had we carried out our experiments in smaller volumes to fit in the plate reader, the total amount of time needed would have also limited the breadth of our study. This is because each community experiment occupied two 96-well plates and spanned the course of 288 hours (12 days), meaning that continuous monitoring of the changes in growth rates of a community would have required at least 24 days per community with one plate reader.

Having considered these challenges, we decided to limit our current study to final yields and taxonomic compositions. However, current work in the lab is focused on studying growth rates using shorter-term experiments with fewer carbon sources and simpler communities. These are being performed using continuous plate-reader monitoring, and we hope they will shed light on microbial growth patterns (e.g. diauxie) on multiple resources.

MINOR COMMENTS

4. Line 20: I am not a huge fan of the term 'molecular complexity'. The term 'molecular' is ambiguous in this context. I would prefer the term 'nutrient complexity'.

We have changed the term according to the reviewer's suggestion.

5. Line 165: I would be careful with the term "co-existence". An unambiguous demonstration of co-existence typically requires a reciprocal invasion experiment.

We have changed our wording to "Our results showed that, while multiple organisms could generally **persist** in environments with single nutrients or with multiple different types of nutrients..." (Line 307).

Reviewer #4 (Remarks to the Author):

The authors present a systematic study of microbial resource competitions where sets of specific species are grown on sets of specific carbon sources. The study is admirable for the care taken to allow for comparisons between outcomes, e.g. by providing a fixed amount of carbon for each competition and combining species and carbon sources in ways that allow for inference of epistatic interactions. There are several results worthy of note. A clear result is that small numbers of species (with no "generalist" species) often fail to achieve maximal final biomass. The authors attribute this to a mismatch between the provided nutrients and the repertoire of nutrients consumable by the species present. While this outcome is not particularly surprising, it has not been definitively demonstrated before to my knowledge. A more interesting observation is the prevalence of negative epistasis for final diversity of species upon mixing two sets of resources. Overall, I found the work to be a solid contribution to address an important question, namely how the variety of nutrients influences microbial diversity. That said, my enthusiasm for the work would be substantially higher if the authors could provide more mechanistic insight into the origin(s) of the observed negative epistasis. I believe this could be accomplished by improvements to the modeling without requiring additional experiments, as discussed below.

We thank the reviewer for their positive evaluation of our work and for their comments.

Major issues:

1. Modeling. In my view, the current version of the model misses a critical aspect of the experiments - specifically the serial dilution protocol. Resource competition in a chemostat and in serial dilution can be quite different because in the former resources are constant, whereas there is a time course of nutrient availability in the latter. I believe that some of the results, including the negative epistasis among nutrients, may only be understandable by explicitly considering the nutrient time course. For example, a recent study by Erez and coworkers (eLife 2020) identified an early bird effect in which species that grew best early in serial dilution outcompeted late growers due to the early increase of the former's population. This would seem to be consistent with the observed negative epistasis: if a species has an early growth advantage on either of the nutrient sets, it would also be an early bird on the mixed

set, also leading to reduced diversity. The authors should reformulate their model to better match the serial dilution protocol of the experiments. This would be a major improvement to the manuscript, and would provide valuable insight into the intriguing observation of negative epistasis. Optimistically, it might even be possible to provide mechanistic insight into the double power law behavior seen in Fig. S14A.

We thank the reviewer for this excellent point. While our modeling did employ average dilution and resource replenishment rates that were in line with our experiment, they did not specifically simulate the serial dilution protocol we used. We fully agree with the reviewer that our predictions of epistasis could be better compared to experiments if they employed the same protocol, so we have redesigned our consumer resource model to explicitly model our experimental process (see Methods, Line 1086 in the document with tracked changes). We have re-run all CRM simulations with this updated protocol, and have regenerated the relevant figures that depend on these modeling results. As a representative example, we would first like to highlight Supplementary Figure 26, which shows *in silico* analogs of com3 and com4 growing on multiple resources. This figure shows the effects of dilutions every 48 hours on organism and resource abundances.

Integrating serial dilutions at 48-hour time points also allowed us to better understand the distributions of yield epistasis E_Y that we observed experimentally. Specifically, when our model was parameterized with the resource utilization profiles of our organisms, it was able to recapitulate the increases in yield displayed by com3 and com4 (Supplementary Figure 5h, l). We had previously been unable to recapitulate these yield increases with our chemostat CRM, which suggests that the specific experimental dilution regime is an important factor in determining the steady-state yield of the community (Line 598).

These results support our hypothesis that a greater diversity of resources leads to a higher probability that specialist organisms will be able to utilize any resource, leading to higher average yields (Line 263). Though our model also predicted increases in yield for our 13-species communities (Supplementary Figure 5d), the magnitude of these increases was very small compared to those of our 3- and 4-species communities. These differences further support an interplay between

community species richness and resource utilization capabilities in determining growth yields and potential nonlinearities.

In addition to allowing us to better contextualize the distribution of yield epistasis, incorporating serial dilutions into our models also provided valuable insight on our observations of taxonomic diversity. In this case, however, our models predicted a stronger negative skewing of species richness and Shannon entropy epistasis when incorporating generalists and specialists (CRM-B, Figure 3c, d, f, g). These resulted in an underprediction of the experimentally-observed diversity metrics compared to our previous chemostat model. Nonetheless, our serial dilution version of CRM-B still correctly predicted a negatively-skewed distribution of E_S and E_H compared to CRM-A, reinforcing our observation that uneven resource use profiles contribute to lower-than-expected degrees of taxonomic diversity.

We thank the reviewer again for this suggestion, and have also updated our scripts available on Github (github.com/segrelab/EnvironmentalComplexity) to include the serial dilution modeling protocol.

2. Discussion. The discussion more or less just repeats the highlights of the Results, and barely puts the work in any broader context. Additional insights from modeling would help here, as would consideration of relevant previous studies. Ecologists have been studying the relationship between ecosystem yield/function and diversity for a long time and it is still an area of intense study. It would be helpful to discuss how the current study fits into this spectrum, e.g. Langenheder et al. PLOS ONE 2010 as just one example. This comes back again to the minimal consideration of mechanism in the current work. How could different results regarding richness vs. nutrient diversity reflect different mechanisms at work?

We thank the reviewer for this comment, and for the opportunity to connect our study to broader ecological research. We have significantly expanded our Discussion section, which now includes commentary on the similarities between our yield epistasis metric and the concept of over/underyielding in the biodiversity-ecosystem functioning (BEF) literature (Line 517). We have also addressed the results of our expanded consumer resource modeling in greater detail, which allowed us to

comment further on some of the possible mechanisms underlying our observations of epistasis.

Minor issues:

3. This phrase in the abstract was particularly opaque until I had read the paper: "...rules that govern how communities respond nonlinearly to the coupling of different nutrient sets". I'm sure this can be rephrased to provide more insight to someone reading through the abstract.

We have reworded the abstract to more clearly define the scope of our study and its results.

4. Some additional value would be added to Fig. 3a by clustering similar outcomes, at least for the single nutrient cases.

We had previously generated a clustering of outcomes (Supplementary Figure 14a), which revealed four main classes of taxonomic distributions. This figure only displayed the clustering diagram itself, so we have generated a new representation of the results in Fig. 3a ordered according to the clustering results (Supplementary Figure 14b). We thank the reviewer for this suggestion as we think it allows for a much clearer interpretation of the clustering analysis.

5. For the analysis of negative epistasis between nutrient sets in Figs. 3f,g, rather than only taking as a baseline prediction the larger parent diversity, it would be worth also showing the results taking as a baseline the smaller parent diversity.

Thank you for bringing up this point. We did consider alternative baselines, as we found while examining the literature on genetic epistasis that an arbitrary (but informed) choice is generally needed to establish a null hypothesis (PMID: 20026678 PMID: 18305163). These often depend on the specific questions or applications involved, for which one definition might be more appropriate than the other. In our case, we have two primary reasons for choosing to use the larger parent diversity:

1. It aligns with the intuitive biological expectation that if certain organisms grew on two separate sets of resources, they should grow on the combination of that set.
2. After calculating E_S and E_H metrics using the larger parent diversity, we saw that our null model was centered at zero, making it an appropriate baseline to which to compare deviations.

Having considered these points, we are grateful for the reviewer's suggestion but feel that it may be confusing to the reader to present multiple definitions of epistasis in the text. Nonetheless, we added a sentence in the Discussion (Line 583) that references previous work in genetic epistasis and clarifies how multiple definitions may offer alternative perspectives on our observations. These may themselves form the basis for an expanded series of quantitative evaluations of nonlinearities in microbial ecosystems.

6. On lines 223 and 224, the same inequality is written for both positive and negative epistasis, the second inequality should be flipped.

Thank you, this has been corrected (Line 443).

7. It's unclear what the authors have tested with the t-tests in Fig. 2d. They state that the distributions are centered at zero, but then show their means are statistically different from a zero centered distribution. This should be clarified.

We have rephrased our figure caption to clarify the exact comparisons being made. Briefly, the distributions of E_Y for each of the three communities is compared to those of statistical ensembles of modeled communities containing the same number of initial organisms (e.g. the com3 distribution is compared to that of a 3-species *in silico* community).

8. There is a repeated typo in the Monod consumption formula: it should be a plus sign in the denominator, so $R/(k + R)$.

Thank you for pointing out this typo. The Monod formula had been correctly implemented throughout our modeling, but erroneously transcribed into the manuscript. We have corrected all instances of this term.

Reviewers' Comments:

Reviewer #1:

None

Reviewer #2:

Remarks to the Author:

The authors have addressed my points in their revision, and I have no further comments. I think this paper will be a nice addition to the literature.

Reviewer #3:

Remarks to the Author:

REVIEWER 3

The reviewers have addressed all of my major concerns and comments. I continue to believe that it is a timely manuscript and sets the stage for a large volume of new and exciting studies.

David Johnson (Eawag)

Reviewer #4:

Remarks to the Author:

The authors have addressed all of my concerns in the revised manuscript, and I am happy to recommend publication in Nature Communications.